

# Characterizing Wind Gusts in Complex Terrain

Frederick Letson[1,2], Rebecca J. Barthelmie[2], Weifei Hu[1,2], Sara C. Pryor[1]

[1]Department of Earth and Atmospheric Sciences, Cornell University, Ithaca, New York
[2]Sibley School of Mechanical and Aerospace Engineering, Cornell University, Ithaca, New York

*Correspondence to*: Frederick Letson (fl368@cornell.edu) and Sara C. Pryor (sp2279@cornell.edu)

**Abstract**

Wind gusts are a key driver of aerodynamic loading, especially for tall structures such a bridges and wind turbines. However,
gust characteristics in complex terrain are not well understood and common approximations used to describe wind gust behavior
may not be appropriate at heights relevant to wind turbines and other structures. Data collected in the Perdigão experiment are
analyzed herein to provide a foundation for improved wind gust characterization and process-level understanding of flow
intermittency in complex terrain. High-resolution observations from sonic anemometers and vertically pointing Doppler lidars
are used to conduct a detailed study of gust characteristics with a specific focus on the parent distributions of nine gust
parameters (that describe velocity, time and length scales), their joint distributions, height variation and coherence in the vertical
and horizontal planes. Best-fit distributional forms for varying gust properties show good agreement with those from previous
experiments in moderately complex terrain but generate non-conservative estimates of the gust properties that are of key
importance to structural loading. Probability distributions of gust magnitude derived from vertical pointing Doppler lidars exhibit
good agreement with estimates from sonic anemometers despite differences arising from volumetric averaging and the terrain
complexity. Wind speed coherence functions during gusty periods (which are important to structural wind loading) are similar to
less complex sites for small vertical displacements (10 to 40 m), but do not exhibit an exponential form for larger horizontal
displacements (800 to 1500 m).

## 1    Introduction and objectives

Topographic channeling or enhancement of the near-surface flow can lead to local increases in wind speed (Wagenbrenner et al.,
2016) and hence enhance the wind resource (Clifton et al., 2014;Barthelmie et al., 2016;Jubayer and Hangan, 2018). Terrain
inhomogeneity also induces complex flow conditions (Wood, 2000) particularly in the presence of vegetation (Suomi et al.,
2013) that have implications for wind loading on structures, pollutant dispersion, wildfire propagation and wind turbine siting
and operation (Sanz Rodrigo et al., 2017;Wagenbrenner et al., 2016;Butler et al., 2015). Key features of flow in complex terrain
include: Thermo-topographic flows arising from differential heating (Rucker et al., 2008;Rotach and Zardi, 2007), and lee-side
vortices that develop parallel to mountain ridges (Grubišić et al., 2008). Regions with complex topography and land cover
heterogeneity also tend to experience more frequent and stronger wind gusts (coherent short-term wind speed maxima) (Letson
et al., 2018;Earl et al., 2017;Sheridan, 2011;Hasager et al., 2003) due in part to:

1.  Terrain-induced alteration of the structure of mesoscale convective systems and thus the downdrafts and wind gusts
generated therefrom (Markowski and Dotzek, 2011).

2.  Generation of small amplitude mountain waves in stably stratified air that can cause strong and gusty downslope winds
when the flow becomes supercritical and these waves 'break' (see detailed discussion in (Durran, 1990) and
(Hertenstein and Kuettner, 2005)).



Wind gusts represent an important source of structural engineering loads for tall buildings, towers, bridges and wind turbines (Solari, 1987;IEC, 2005;Cheynet et al., 2016), and are known to be of larger magnitude in complex terrain due in part to the factors listed above (Tieleman, 1992;Verheij et al., 1992). A number of numerical wind flow models have been developed for application at high spatial resolution over complex terrain, but model evaluation has been severely constrained by the lack of

suitable observational data (Butler et al., 2015;Bechmann et al., 2011;Berg et al., 2011;Suomi and Vihma, 2018). Further, most past research on flow intermittency has focused on the intensity (i.e. magnitude) of wind gusts and has employed measurements from 10 m a.g.l. (e.g. (Vickery and Skerlj, 2005)). Thus, there is a need to advance understanding of the spatio-temporal coherence of wind gusts at heights above 10 m a.g.l., in complex terrain (Belu and Koracin, 2013;Mouzakis et al., 1999), and for better characterization of both (i) the height variation of gust properties (Suomi et al., 2013) and (ii) to characterize additional

descriptors of wind gusts such as gust rise times and length scales since these properties also contribute to the wind-excited structural response (Solari, 2014;Frost and Turner, 1982) and fatigue loading on wind turbines (Chamorro et al., 2015;Hu et al., 2016).

Herein we address these research needs using data collected during January – July 2017 at a site in eastern Portugal near Perdigão (Figure 1a). Two parallel ridges running from northwest to southeast and separated by 1.4 km dominate the local

topography in the study area. These ridges stand 300 to 350 m above the surrounding terrain, and approximately 175 m above the valley located between them (Figure 1b). This location was the focus of a measurement campaign during which over 50 meteorological masts were deployed over an area of a few square km (Mann et al., 2017). The data collected in the Perdigão experiment and employed herein to characterize flow behavior at heights relevant to wind turbine selection, operation and micro-siting with a specific focus on wind gusts are: High-frequency (18 Hz) 3-D wind measurements from Gill WindMaster Pro sonic

anemometers deployed on the nine tallest of the meteorological masts (that extended to heights (z) above 50 m a.g.l.) and horizontal wind speeds from two vertically-pointing conically scanning (ZephIR) Doppler lidars (locations of these instruments are shown in Figure 1b and Table 1, details of the measurement technologies are given in section 2).

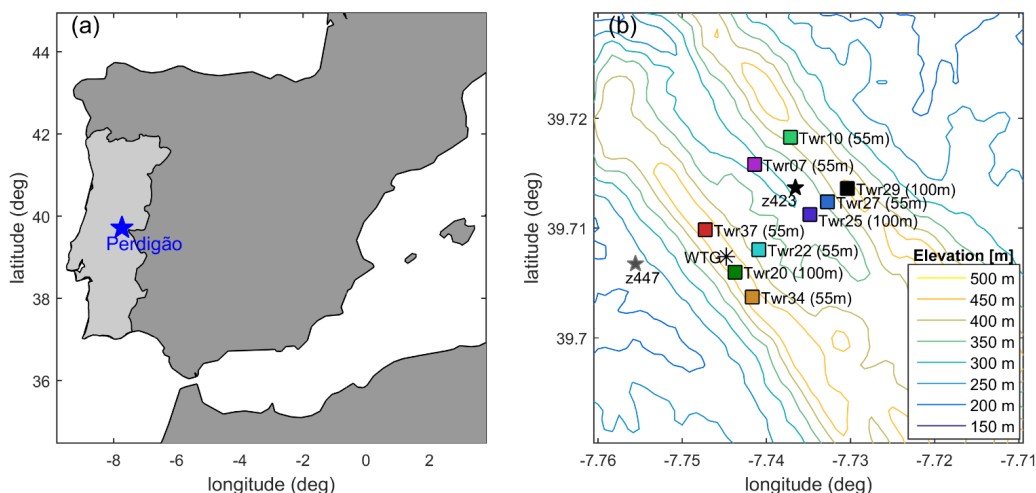

**Figure 1: (a) Location of the Perdigão measurement site in Portugal. (b) Overview of the tower and ZephIR lidar locations and site
topography (measured by the Shuttle Radar Topography Mission (Farr et al., 2007)). The height of each tower is shown in
parentheses. The nine colors shown in this figure are used to distinguish the nine towers throughout the paper. A single wind turbine is
located on the SW ridge (denoted by WTG *).**

The objectives of the current study are as follows:





- Evaluate the degree to which the best-fit probability distributions to various descriptors of wind gusts (e.g. intensity (gust factors, gust amplitude and peak factor), temporal scale (rise and lapse time, and duration), and length scale) as advanced by (Hu et al., 2018), are generalizable across terrain types. The resulting parametric descriptions of gust properties are potentially of utility to the engineering community because they; permit estimation of extreme values (IEC, 2005;ASCE, 1998) (e.g. using Rice theory (Gomes and Vickery, 1977)), facilitate development of joint distributions of gust properties, allow characterization of gusts that contribute to structural fatigue, and are used with design standards (for example, extreme gusts are modeled in wind turbine design standards based on mean wind speeds and turbulence intensity (IEC, 2005)). They are potentially also of use within the meteorological community since they could afford a methodology for downscaling of wind gusts in either weather forecasting (Friederichs and Thorarinsdottir, 2012;Suomi and Vihma, 2018) or climate downscaling contexts (Cheng et al., 2014). Further, fluctuating wind loads on engineering structures requires estimates of multiple components of the flow, including characteristics that have previously received relatively little attention (e.g. the shape of wind gusts) (Mücke et al., 2011;Suomi et al., 2013). Various parametric distributions are evaluated in terms of their goodness-of-fit to the empirical data, and their accuracy at the distribution tails, and are used to develop joint probability distributions of different gust properties at a single location and of the same gust property across space (where the latter can be used to develop bivariate extreme value Copulas (Bonazzi et al., 2012)). Where possible the distributional forms for each gust parameter are compared with previous work in flat or moderately complex terrain (Morgan et al., 2011;Cheng and Bierbooms, 2001;Friederichs and Thorarinsdottir, 2012;Hu et al., 2018).

- Quantify the dependence of different descriptors of wind gusts on measurement height $(z)$. Data from comparatively flat terrain show evidence that the characteristics of wind gusts, and particularly gust factors and gust durations, vary systematically with height (Román, 2017;Suomi et al., 2015;Suomi et al., 2013;Ashcroft, 1994). We seek to describe the magnitude and nature of this variability with height in complex terrain by conditionally sampling gust properties as derived from the sonic anemometers deployed on the meteorological masts and as determined from vertically pointing ZephIR lidars.

- Characterize power spectra of wind speeds from sonic anemometers and ZephIR lidars at different heights. These power spectra are used to determine how the presence of wind gusts affects their shape (Hu et al., 2018), and to derive first order estimates of the so-called reverse height (i.e. height above ground at which surface-driven processes cease to dominate scales of variability) using the amount of variance expressed at the diurnal timescale (Larsén et al., 2018;Troen and Lundtang Petersen, 1989). In the near-surface levels surface-driven processes produce the diurnal peak in the power spectrum of wind speeds, while aloft it is primarily the product of pressure perturbations deriving from the atmospheric tide (Larsén et al., 2018). At intermediate heights there is a relative minimum in the amount of variance expressed at periods $\approx 1$ day. At these heights the first-order effect of the surface heat-flux modulations vanishes (Larsén et al., 2018), and thus it may provide an estimate of the height at which surface-driven processes cease to dominate scales of variability.

- Quantify the dependence of wind gust characteristics on atmospheric conditions; specifically stability, wind direction and turbulence intensity (Barthelmie et al., 2016;Hu et al., 2018). Previous work has shown that gust factors are strongly and directly related to turbulence intensities (Ashcroft, 1994;Greenway, 1979;Hu et al., 2018), and that turbulent kinetic energy (and hence the potential for gusts) is enhanced downstream of obstacles (Jubayer and Hangan, 2018). Thus, wind gust properties at the nine towers are conditionally sampled by wind direction, stability class and by turbulence intensity.

- Quantify spatial coherence in flow properties particularly wind gusts. The physical scales of wind gusts are critically important to loading on structures (Solari, 1987;Hui et al., 2009;Bos et al., 2016), and the potential for gusts to remain coherent as they propagate through a wind farm has implications for power quality and grid management (Sørensen et al.,



2002;Vigueras-Rodríguez et al., 2012). The frequency characteristics of longitudinal wind speed are characterized using spectral analysis of output from individual sonic anemometers and coherence functions between pairs of sonic anemometers during gusty periods. Horizontal coherence functions between sonic anemometers on different masts and thus displaced by distance of 100's of meters are used to describe the degree to which wind gusts are simultaneously present (i.e. within an X

minute time window) across the study domain.

## 2   Data

High precision estimates of the terrain elevation and canopy height were derived from aerial laser scans performed by helicopter. The x, y and z positions of the maximum backscatter form a point cloud and are processed to derive terrain elevation and height of the canopy (Floors et al., 2018;Boudreault et al., 2015). Mean maximum canopy heights derived from these data for 50 m by

50 m grid cells centered on each of the nine meteorological masts and the ZephIR lidars are below 7 m (Table 1).

The primary data set analyzed herein comprises 18 Hz, 3-D wind components and sonic virtual temperature as measured by Gill WindMaster Pro sonic anemometers deployed on these meteorological masts (Table 1 and Figure 1) at heights above the surrounding vegetation. The three tallest towers have seven measurement heights (z) each extending from 10 to 100 m a.g.l. and the remaining six towers have five measurement heights each extending from 10 m to 55 m. The 18 Hz signals from each sonic

anemometer are subject to coordinate rotation (including corrections for the boom alignment), and despiking using a five-sigma filter in each 10 minute period. In order to ensure that our characterization of spatial variations in wind gust characteristics is not biased by differing measurement periods, the current analyses are restricted to days that have complete data records at all anemometers. Data analyzed here represent all 24 hour periods during which all of the 41 sonic anemometers had > 90% of all 18 Hz signals present in > 99% of 10 minute periods (143 out of 144, 10 minute periods in each day). Analyses of gust

parameters based on data from the ZephIR Doppler lidar measurements are also for the same 64, 24 hour periods.

Table 1: Locations and measurement heights of each meteorological mast (Tower) and each ZephIR lidar. Measurement heights referred to in this paper as '60 m' are shown in bold. Reference tower (Tower 29) is emphasized by italics and underlining. Tower base elevations are given in m above sea level (a.s.l.) and mean vegetation height is calculated from aerial laser scans in a 50 m square cell surrounding each tower/ZephIR lidar. Definitions used to conditionally sample the towers as valley or ridge are also shown along with

a parenthetical statement of their location on the northeast (NE) or southwest (SW) ridge. Valley towers are those with elevations below 400 m a.s.l.

| Tower # | Measurement heights [m a.g.l.] | | | | | | | Tower elevation [m a.s.l.] | Latitude [°N] | Longitude [°W] | Location | Mean canopy height [m] |
|---|---|---|---|---|---|---|---|---|---|---|---|---|
| 7 | 10 | 20 | 30 | 40 | **55** | | | 290 | 39.7158 | 7.7414 | valley | 3.3 |
| 10 | 10 | 20 | 30 | 40 | **55** | | | 413 | 39.7183 | 7.7372 | ridge (NE) | 3.4 |
| 20 | 10 | 20 | 30 | 40 | **60** | 78 | 100 | 465 | 39.7060 | 7.7437 | ridge (SW) | 1.5 |
| 22 | 10 | 20 | 30 | 40 | **55** | | | 385 | 39.7080 | 7.7409 | valley | 3.8 |
| 25 | 10 | 20 | 30 | 40 | **60** | 80 | 100 | 309 | 39.7112 | 7.7348 | valley | 5.4 |
| 27 | 10 | 20 | 30 | 40 | **55** | | | 359 | 39.7124 | 7.7327 | valley | 6.7 |
| _29_ | _10_ | _20_ | _30_ | _40_ | _**60**_ | _80_ | _100_ | _450_ | _39.7136_ | _7.7304_ | _ridge (NE)_ | _3.6_ |
| 34 | 10 | 20 | 30 | 40 | **55** | | | 468 | 39.7037 | 7.7417 | ridge (SW) | 2.9 |
| 37 | 10 | 20 | 30 | 40 | **55** | | | 474 | 39.7098 | 7.7473 | ridge (SW) | 1.9 |
| ZephIR z423 | 20:20:200 | | | | | | | 310 | 39.7137 | 7.7366 | valley | 0.9 |
| ZephIR z477 | 20:20:200 | | | | | | | 236 | 39.7067 | 7.7556 | west of SW ridge | 2.2 |

Tower 29, the 100 m tower on the northeast ridge (Figure 1), is used herein as a reference tower to represent pseudo free-stream flow and characterize the prevailing atmospheric stability because of the prevalence of northeasterly flow during the field

experiment (Figure 2). Measurements from this meteorological mast indicate a high frequency of flow perpendicular to the




ridges. Wind directions between 30° and 60° (i.e. within ± 15° of northeasterly) occurred during 20% of 10 minute periods while wind directions between 210° and 240° (i.e. within ± 15° of southwesterly) occurred during 14% of 10 minute periods (Figure 2c). At Tower 29, the mean 10 minute wind speed at 60 m is 5.0 ms⁻¹, and the mean turbulence intensity for mean wind speeds > 3 ms⁻¹ is 0.14 (Figure 2a,c). The greatest height represented at all nine meteorological masts is 55 or 60 m. Thus, this height is

used to compare wind conditions across study domain and is referred to herein as the 60 m measurement height for brevity. In some of the following analyses the meteorological masts (towers) are classified as 'ridge' or 'valley' where the former group have base elevations above 400 m a.s.l., and the latter are at elevations below that level (see Table 1).

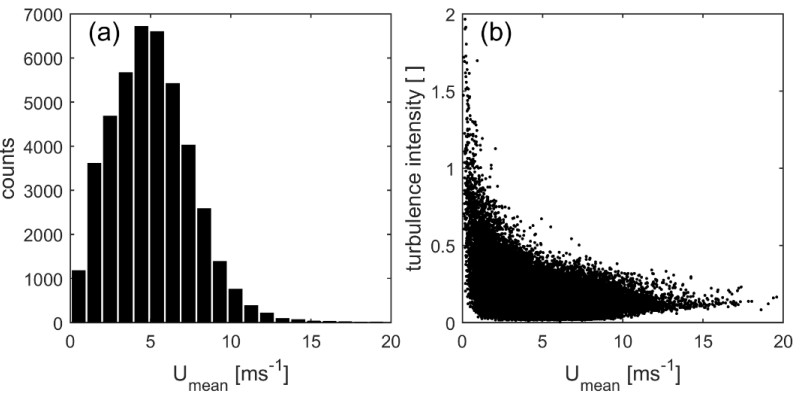
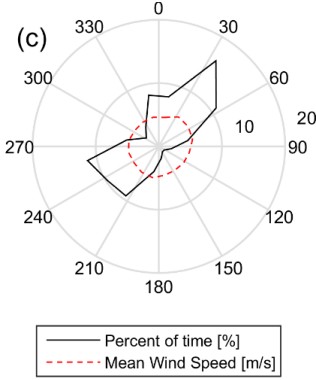

**Figure 2: Overview of wind conditions at Perdigão. (a) 10 minute mean wind speed distribution at 60 m a.g.l. on Tower 29. (b) 10**
**minute turbulence intensities versus mean wind speed at 60 m a.g.l. on Tower 29. (c) Wind rose and mean wind speed from data**
**collected at 60 m a.g.l. on Tower 29.**

Some analyses reported herein employ measurements from two ZephIR Doppler lidars (z447 and z423). One was deployed in the central valley 311 m from meteorological mast 25, and one to the west of the SW ridge (Figure 1b; locations in Table 1). The ZephIR series 300 units use a wavelength of 1.575 μm, an operating frequency of 50 Hz, and have a probe length of 0.07 m at 10

m and 7.7 m at 100 m. They were configured to measure horizontal wind speeds at 10 measurement heights (20 to 200 m a.g.l. at 20 m intervals), sampling at each height once every 17 s. Conically scanning Doppler lidars (such as the ZephIR 300 continuous wave lidars (Smith et al., 2006) used herein) can be used to retrieve relatively high frequency (and time averaged) horizontal wind speeds at multiple heights (in the case of the ZephIR 300 instruments to a height of 200 m) above the instrument, and have been subject to extensive field validation (Gottschall et al., 2012). However, these Doppler lidars assume the flow to be

homogeneous across the scanning volume (the ZephIR use a scanning cone angle of ±15°) in order to infer the horizontal wind speed. This assumption is not fulfilled in complex terrain due leading to increased uncertainty and potentially error in retrieved horizontal wind speeds (Bingöl et al., 2009). Further, the volumetric averaging inherent in use of lidars means the wind speeds are not directly equivalent to those from in situ anemometry. Nevertheless, vertically pointing Doppler lidars offer the potential to quantify wind gusts at heights above those possible from sonic anemometers deployed on meteorological masts (Suomi et al.,

2017), and are increasingly being adopted by the wind energy research and operations communities (IEA Wind Task-32 Lidar for Wind Energy Deployment) (Clifton et al., 2018). Herein, we evaluate the degree to which the probability distribution of gust amplitudes and GF derived from the maximum of the disjunct measurements sampled at each of 10 heights (effective duration of approximately 2 sec) in each 10 minute period correspond to those from the 18 Hz data from the sonic anemometers. We also use time series of 10 minute mean wind speeds from each height (up to 200 m a.g.l.) from the ZephIR lidars to investigate the

reversal height (implied by the magnitude of the diurnal peak, estimated across the frequency range; $3 \times 10^{-6}$ to $2 \times 10^{-5}$ Hz) and to





compute coherence functions. For this purpose, the longest continuous (any/all data gaps < 1 day in duration) data period from each ZephIR lidar is used (91 days of data at z423 and 142 days at z447).

## 3    Methods

Section 3.1 provides definitions used herein and outlines methods used in the conditional sampling, while section 3.2 briefly describes the methods used in the spectral and coherence analyses.

### 3.1    Wind gust descriptors and their probability distributions

The following definitions are used herein:

1.  **$U_{mean}$ [ms$^{-1}$]**: 10 minute mean longitudinal wind speed.
2.  **Wind Direction [°]**: 10 minute mean wind direction.
3.  **Gust magnitude ($U_{gust}$) [ms$^{-1}$]**: Maximum value of a 3 s moving average longitudinal wind speed during a 10 minute period.
4.  **Gust amplitude ($a_{gust}$) [ms$^{-1}$]**: Deviation of the gust wind speed from the mean: $U_{gust} - U_{mean}$
5.  **Peak Factor ($k_{peak}$) [ ]**: 3 s gust amplitude ($a_{gust}$) normalized by the standard deviation (σ) of the 18 Hz longitudinal wind speed during the 10 minute period.
6.  **Gust factor (GF) [ ]**: Ratio between the 3 s gust magnitude and the 10 minute mean wind speed: $U_{gust} / U_{mean}$
7.  **Rise time ($t_{rise}$) [s]**: Time elapsed between the occurrence of the maximum 3 s wind speed ($U_{gust}$) and the immediately preceding local minimum in the 3 s moving average that is below $U_{mean}$.
8.  **Lapse time ($t_{lapse}$) [s]**: Time elapsed between the occurrence of $U_{gust}$ and the next local minimum in the 3 s moving average that is below $U_{mean}$.
9.  **Gust duration, ($t_{gust}$) [s]:** $t_{gust} = t_{rise} + t_{lapse}$
10. **Gust length scale ($L_{gust}$) [m]**: An estimate of the physical extent of a wind gust, defined as the integral of the 3 s moving average of longitudinal wind speed during the duration of the gust
11. **Turbulence intensity (TI) [ ]**: Standard deviation (σ) of the 18 Hz longitudinal wind speeds during the 10 minute period divided by the 10 minute mean wind speed, σ/$U_{mean}$
12. **Stability class:** Five classes denoting atmospheric stability based on Monin-Obukhov length ($L$):

$$L = \frac{u_*^3}{k(\frac{g}{\theta})\overline{w'\theta'}} \tag{1}$$

Where κ is the von Karman constant, $u_*$ is the friction velocity, g is the acceleration due to gravity, and w' and θ' are the fluctuating components of vertical velocity and sonic virtual temperature respectively. It is acknowledged that the surface similarity theory that underpins use of $L$ as a stability parameter derives from measurements in flat terrain and within the surface layer (Monin and Obukhov, 1954). Thus, $L$ as computed based on measurements at 60 m on the reference tower (29) is used to conditionally sample the gust properties based on broad stability classes from (Barthelmie, 1999) wherein; $0 < L < 200$ m indicates very stable conditions, $200 < L < 1000$ m is stable, $|L| > 1000$ m is neutral, $-1000 < L < -200$ m is unstable and $-100 < L < 0$ m is used to indicate very unstable conditions.

Following (Hu et al., 2018) four 2-parameter probability distribution types are fitted to the gust parameters (1 and 3-10, above) as derived from time series from sonic anemometers on all meteorological masts, for all 10 minute periods when $U_{mean} > 3$ ms$^{-1}$. These four distribution types and their probability density functions are as follows:



1. Weibull

$$f(x|a,b) = \frac{b}{a}\left(\frac{x}{a}\right)^{b-1} exp\left[-\left(\frac{x}{a}\right)^b\right] \tag{2}$$

2. Log-logistic

$$f(x|a,b) = \frac{exp\left[\frac{\ln(x)-a}{b}\right]}{bx\left\{1+exp\left[\frac{\ln(x)-a}{b}\right]\right\}^2} \tag{3}$$

3. Lognormal

$$f(x|a,b) = \frac{1}{xb\sqrt{2\pi}}exp\left[\frac{-(\ln x-a)^2}{2b}\right] \tag{4}$$

4. Gamma

$$f(x|a,b) = \frac{1}{b^a \Gamma(a)}x^{a-1}exp\left(-\frac{x}{b}\right) \tag{5}$$

Where $\Gamma$ is the gamma function, x is the random variable being described, and a and b are the distribution parameters.

Distributions are fitted to each gust parameter using maximum likelihood estimation (MLE), and best fit distribution types are determined using negative-log-likelihood (NLL) values (Hogg et al., 2005). Since two or more distributional forms may exhibit relatively good fits to the empirical distributions we also note results wherein a second distribution type exhibits equivalent NLL values (i.e. those within 0.1% of the 'best fit'). The tails of probability distributions are typically of greatest importance to wind loading (e.g. turbine design and control systems (IEC, 2005)) and are not always well described by distributional forms that best represent the body of the distributions (Friederichs and Thorarinsdottir, 2012). Thus, the effectiveness of each distribution type in representing the 99[th] percentile gust magnitude and gust amplitude, and the 1[st] percentile rise time is evaluated by comparing the parametric estimate derived from the fitted distribution to the empirically derived percentile value.

Once distributional forms for individual gust properties have been derived they are used to construct joint distribution of gust parameters using a general method that converts gust parameters following any type of distribution to the standard Gaussian domain and generates the joint distribution of the transformed gust parameters. For gust parameters following Weibull distribution with the probability density function (PDF, Eq (2)) the transformation to a Gaussian form is realized using the following explicit equation:

$$U = -\Phi^{-1}\left[exp\left(-\left(\frac{X}{a}\right)^b\right)\right] \tag{6}$$

where $a$ and $b$ are the two PDF parameters (scale parameter and shape parameter, respectively) calculated using the MLE method. $X$ and $U$ represent the original random variable following Weibull distribution and the transformed random variable following standard Gaussian distribution, respectively. For gust parameters following a lognormal distribution (Eq (4)), the explicit transformation equation is expressed as

$$U = \frac{\ln(X)-a}{b} \tag{7}$$

For gust parameters that follow a Gamma distribution (e.g., gust length scale at tower 29), there is no explicit transformation equation. Thus, the gust parameters are empirically transformed to standard Gaussian variables using:

$$U = \Phi^{-1}\left[F(X)\right] \tag{8}$$

where $F(X)$ is the empirical cumulative distribution function (CDF) of random variable $X$ and $\Phi^{-1}$ is the inverse CDF of standard Gaussian random variable.

After the gust parameters are transformed to standard normal variables, 2-D elliptical contours are computed that enclose a specified percent of transformed data using the fact that the sum of squared Gaussian random variables follows a Chi-Square





distribution. The orientation angle, major axis, and minor axis of the ellipse are calculated from eigenvalues and eigenvectors of the covariance matrix of each pair of transformed gust parameters (Wilks, 2011). The aspect ratio between the lengths of the major axis and the minor axis of the ellipse indicate the correlation between the gust parameters. Herein we report joint distributions of a single gust parameter at two heights from the same tower and joint distributions of two different gust

parameters at the same height from two towers.

In numerical weather prediction (NWP) models, wind gusts are generally sub-grid scale and thus are estimated using parameterizations. In the simplest case, the peak factor ($k_{peak}$) is assumed to be a constant factor of 1.7 (Woetmann Nielsen and Petersen, 2001). An approximation, derived using measurements between 8 and 80 m a.gl. near a lake and within a city (Wieringa, 1973), that describes $k_{peak}$ (as measured over an averaging period of t) as:

$$k_{peak} = 1 + \frac{1.42 + 0.3013\, ln\left(\frac{990}{U_{mean}t} - 4\right)}{ln(\frac{z}{z_0})} \qquad (9)$$

Where $z_0$ is the surface roughness length (a value of 0.5 m is used here). The value of 990 m in the numerator is the wavelength below which an effective majority of the locally-derived turbulent fluctuations are expressed within a 10 minute period, while longer wavelengths derive from mesoscale features (Wieringa, 1973). Estimates from these two approximations are compared to $k_{peak}$ derived from measurements at 60 m height at all 9 towers to establish whether they are conservative in complex terrain.

Conditional sampling is used to explore the functional dependencies of gust properties. In these analyses we classify any 10 minute period that meets two wind intensity criteria: $U_{mean} > 3$ ms$^{-1}$ and gust amplitude ($a_{gust}$) > 4 ms$^{-1}$ as being a 'gust period' and denote all other 10 minute periods as non-gust periods. These thresholds are applied to exclude periods with high GF that occur solely because of low $U_{mean}$, and to ensure the wind gusts represent periods during which typical wind turbines would be operating. The threshold of $a_{gust} > 4$ ms$^{-1}$ is a simple approximation of the gust criteria used in the National Weather Service's

Automated Surface Observation System (ASOS) (NOAA, 2004;Nadolski, 1998), which also results in minimum gust magnitudes of just over 7 ms$^{-1}$. The presence or absence of wind gusts is always determined locally (at a given sonic anemometer). The samples of $U_{mean}$ and gust properties from the different towers and heights do not conform to Gaussian distributions (see examples in Figure 3) thus the central tendency is uniformly described herein using the median. The co-occurrence of wind gusts at pairs of sonic anemometers (sensor 1 and sensor 2) is given as the conditional probability of a gust

occurrence at sensor 2 when a gust occurs at sensor 1 during a given 10 minute period.



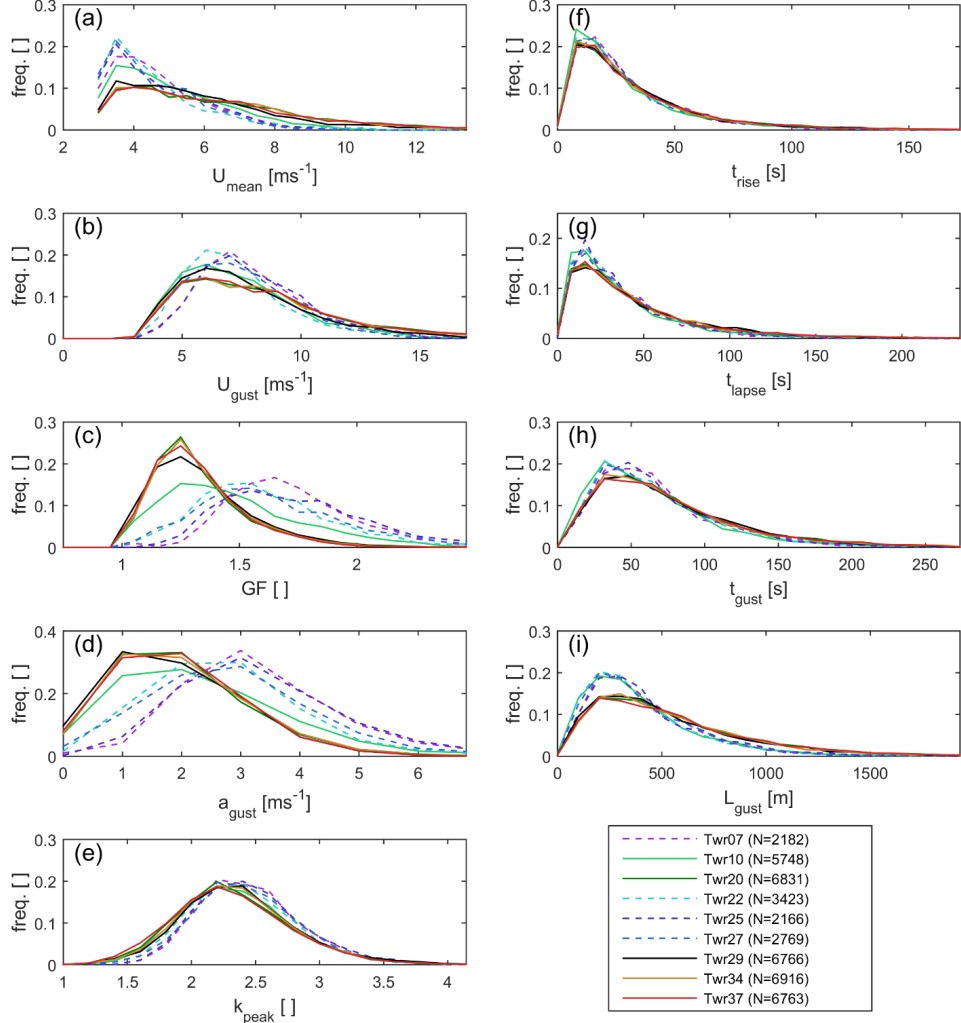

**Figure 3: Distributions of gust parameters at each of the 9 towers (using data collected at 60 m a.g.l) for 10 minute periods with $U_{mean} >$ 3 ms$^{-1}$. Valley towers are shown with dashed lines and ridge towers with solid lines. The frames show the (a) 10 minute mean wind speed ($U_{mean}$), (b) gust magnitude ($U_{gust}$), (c) gust factor (GF), (d) gust amplitude ($a_{gust}$), (e) peak factor ($k_{peak}$) (f) rise time ($t_{rise}$) (g) lapse time ($t_{lapse}$), (h) gust duration ($t_{gust}$) and (i) gust length scale ($L_{gust}$). The legend reports the sample size (N) for 10 minute periods used to construct the empirical distributions for each tower. The towers are denoted by the color scheme introduced in Figure 1.**

### 3.2    Spectra and Coherences

Power spectral densities (PSD) of wind speeds from the sonic anemometers and the ZephIR lidars are calculated using Welch's method (Welch, 1967). For the sonic anemometer data this method is applied to two hour time series of 18 Hz longitudinal wind speed measurements that meet the gust criteria and (separately) those that do not. Spectra are plotted in non-dimensionalized form wherein the power at each frequency is multiplied by the frequency and divided by the variance computed from the 18 Hz time series, and the frequency is multiple by a measurement height (z) of 60 m and the mean wind speed during that 2 hour period. Mean PSD computed for sonic anemometers deployed on all ridge and valley towers are presented for data conditionally sampled by atmospheric stability, turbulence intensity and wind direction.





Spatial relationships of longitudinal wind speeds from the sonic anemometers (and ZephIR lidars) are characterized in the frequency domain using coherence functions, $C_{xy}$, given by the cross-spectral properties:

$$C_{xy}(f) = \frac{|P_{xy}(f)|^2}{P_{xx}(f)P_{yy}(f)} \tag{10}$$

where $f$ is frequency, $P_{xy}(f)$ is the cross-spectral density of x and y, and $P_{xx}$ and $P_{yy}$ are the auto spectral densities of x and y

respectively (Bendat and Piersol, 2011). The normalization means that if two time series are perfectly correlated at a given frequency $C_{xy}(f) = 1$. Wind speed coherence functions are often characterized using a single-parameter exponential decay function, $C_{fit,xy}$ (Solari, 1987):

$$C_{fit,xy}(f) = exp\left(-C\,\frac{f \cdot d}{U_{mean}}\right) \tag{11}$$

where C is the decay coefficient and d is the distance between sensors. A non-dimensional reduced frequency $\left(\frac{f \cdot d}{U_{mean}}\right)$ is used

herein to facilitate comparison of the coherence functions between sonic anemometers deployed on different towers and with previous research (Solari, 1987;Mehrens et al., 2016). Empirical estimates of a coherence function are influenced (and their accuracy limited) by the number of sub-series used in the cross-spectral density calculation. Hence estimates of $C_{xy}(f)$ do not decay to zero but to a coherence floor (Mann, 1994). Thus herein, C values are determined by least squares fitting of Eq. (11) to the empirical coherence function values above 110% of the coherence floor. Previous research has indicated large C values are

most frequently observed in unstable conditions and that coherence decays quickly with reduced frequency (Kristensen and Jensen, 1979). For 10 minute mean wind speeds over water an average value of C = 4.3 has been proposed for horizontal separations of < 5 km (Vigueras-Rodríguez et al., 2012), and the correlation between two measured time series displaced horizontally asymptotes to a constant value (i.e. the coherence floor) at a normalized frequency ≈ 0.36 (Mehrens et al., 2016). Due to the relatively low sampling rate of the ZephIR lidars, coherences of horizontal wind speeds from the ZephIR lidars are

calculated over a 72 hour period with and FFT window length of 256 ($2^8$), while the coherence functions calculated from the sonic anemometer observations are based on 2 hour periods of 18 Hz longitudinal wind speed data, transformed into the frequency domain using an FFT window length of 16384 ($2^{14}$).

Characterization of the height at which the surface characteristics cease to dominate scales of flow in the atmosphere has applications to microscale model verification and validation and is accomplished herein through investigation of the height

dependence of the spectral peak associated with the diurnal time scale ($f = 1$ day[-1]) (Larsén et al., 2018). The magnitude of the diurnal peak, $S_p$, is calculated as a function of height from the PSDs of wind speeds from all ten ZephIR lidar measurement heights and sonic anemometer data from the three tallest towers (20, 25 and 29) as follows (Larsén et al., 2018):

1.  Calculate the PSD using Welch's and the longest complete data period
2.  Perform log-smoothing (35 points per decade) of these PSDs by piecewise cubic interpolation

3.  Fit a linear function to each PSD (log S(f) Vs log(f)), to the data in the range $3 \cdot 10^{-6} < f < 2 \cdot 10^{-5}$ Hz, excluding the value at $F_d$ (frequency = 1 day[-1])
4.  Calculate three parameters:

$S_{reg}(F_d)$: Value of the linear fit at the daily peak frequency, $F_d$

$S(F_d)$: Value of the PSD at $F_d$

$S_p$: Height of the daily peak above the linear background ($S(F_d)$- $S_{reg}(F_d)$)

The reverse height is the height of the minimum value of $S_p$ from each independent measurement and, as described above, is interpreted as the height at which surface-driven processes no longer dominate flow variability.





## 4    Results

### 4.1    Wind Gust Properties and Parameter Distributions

Mean sustained (10 minute) wind speeds ($U_{mean}$) are higher at ridge towers than at towers in the valley. The median values are 5.70 ms$^{-1}$ and 4.26 ms$^{-1}$, respectively (Figure 3a). Tower 10 is an exception to this general pattern because although it is located on the northeast ridge it is sheltered by an area of higher elevation to the north (Figure 1) and thus experiences flow conditions that are, overall, more like the valley towers. Although the full sample of $U_{mean}$ values at all towers is best fit by a Weibull distribution (Figure 2, as at flat sites and offshore (Morgan et al., 2011;Pryor et al., 2004)), when a threshold of 3 ms$^{-1}$ is applied, the resulting samples of $U_{mean}$ values are log-normally distributed at all towers (Table 2).

In accord with a priori expectations, gust amplitudes ($a_{gust} = U_{gust} - U_{mean}$) are higher in the valley than along the ridges (Figure 3d) while $U_{gust}$ (the maximum 3 s moving average in a 10 minute period) exhibits similar values at ridge and valley towers (7.36 ms$^{-1}$ and 7.57 ms$^{-1}$, respectively), with valley towers showing a more peaked distribution and ridge towers exhibiting a longer tail (Figure 3b). The total sample of $U_{gust}$ estimates is best represented by either a Weibull distribution, as commonly used in wind turbine modeling (Cheng and Bierbooms, 2001), or a gamma distribution, which has been used in previous work describe gusts below a canopy (Shaw et al., 1979) (Table 2). Consistent with measurements from moderately complex terrain (Hu et al., 2018), when $U_{gust}$ is conditionally sampled for $U_{mean} > 3$ ms$^{-1}$, it is best fit by a lognormal distribution. Extreme values of $U_{gust}$ (i.e. 99$^{th}$ percentile ($p_{99}$) which ranges from 13.2 to 17.1 ms$^{-1}$ across the towers) are most accurately predicted by the gamma distribution. Parametric estimates of $p_{99}$ are conservative when derived from the log-normal fit, but are biased low from both gamma and Weibull distribution fits to data from sonic anemometers deployed at/close to 60-m a.g.l. on all towers (Figure 4a). Consistent with measurements from moderately complex terrain (Hu et al., 2018), $a_{gust}$ (i.e. the deviation of the maximum 3 s moving average from the 10 minute mean) are best described by a Weibull distribution (Table 2), but 99$^{th}$ percentile values of $a_{gust}$ (5 to 7.8 ms$^{-1}$) estimated from the Weibull parametric fit are also non-conservative. While all other distribution types tend to over-predict the 99$^{th}$ percentile gust amplitude value (Figure 4b), the gamma distribution appears to generate the most representative (but conservative) estimates. Thus, if the upper percentiles of wind gust intensity are of particular interest (e.g. in engineering for wind loading) it may be preferable to use a log-normal distribution to represent $U_{gust}$ and a gamma distribution for gust amplitude ($a_{gust} = U_{gust} - U_{mean}$).

Although gust magnitude and amplitude are useful for determining the loading force exerted by wind gusts, gust factors (GF, i.e. the ratio of the 3-5 s gust magnitude to the sustained wind speed) are frequently used in the meteorological community as a non-dimensional intensity index (Krayer and Marshall, 1992) and are sometimes used for assessment of wind hazards (Deaves, 1993). GF are generally higher in the valley than on the ridge (median GF at valley towers is 27% higher than those from the ridge towers; Figure 3c), consistent with the lower $U_{mean}$ in the valley. GF samples from the 60 m measurement level are best described by the log-logistic distribution (as in a site in moderate terrain complexity (Hu et al., 2018)) or lognormal distributions (as in a sample of sites distributed across the eastern United States (Pryor et al., 2014)) (Table 2).




**Table 2: Best fit distribution types for each gust property (in each 10 minute period when $U_{mean} > 3$ ms$^{-1}$) for a measurement height of 60 m a.g.l. at each tower. Also shown are results from moderately complex terrain (Hu et al., 2018). The distribution types are referred to as 1-4 where 1: Weibull, 2: log-logistic, 3: log-normal, 4: gamma. Distributions types shown in parenthesis represent an equivalently good fit (i.e. those with negative log likelihood of less than 0.1% lower than the best fit distribution).**

|  | Tower | | | | | | | | | Hu et al, 2018 |
|  | 7 | 10 | 20 | 22 | 25 | 27 | 29 | 34 | 37 | |
|---|---|---|---|---|---|---|---|---|---|---|
| $U_{mean}$ | 3 | 3 | 3 | 3 | 3 | 3 | 3 | 3 | 3 | - |
| $U_{gust}$ | 3 | 3 | 3 | 3 | 3 | 3(4) | 3 | 3 | 3 | 3 |
| GF | 2 | 3 | 2 | 3 | 3 | 3 | 3 | 2 | 2 | 2 |
| $a_{gust}$ | 1 | 1 | 1 | 1 | 4 | 1 | 1 | 1 | 1 | 1 |
| $k_{peak}$ | 2 | 2 | 2 | 2 | 3 | 2 | 2 | 2 | 2 | 2 |
| $t_{rise}$ | 3 | 3 | 3 | 3 | 3 | 3 | 3 | 3 | 3 | 3 |
| $t_{lapse}$ | 3 | 3 | 3 | 3 | 3 | 3 | 3 | 3 | 3 | 3 |
| $t_{gust}$ | 3(4) | 3 | 3 | 3 | 3 | 3 | 3(4) | 3 | 4 | 3 |
| $L_{gust}$ | 3(4) | 3 | 4(3) | 3 | 3 | 3(4) | 4 | 4(3) | 4 | 3 |

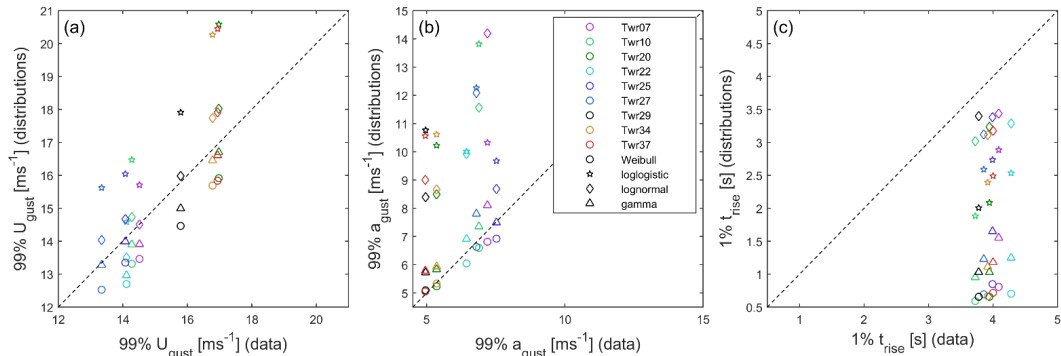

**Figure 4: (a) 99th percentile wind gust magnitude ($U_{gust}$), (b) 99th percentile amplitude ($a_{gust}$) and (c) 1st percentile rise time ($t_{rise}$) as derived directly from the observations at 60 m a.g.l. and estimated from the parametric distributional fits (shown by the different symbols). The towers are denoted by the color scheme introduced in Figure 1. The dashed line indicates 1:1 correspondence.**

Gust rise values are similar at ridge and valley towers, median($t_{rise}$) = 24.3 and 23.0 s, respectively (Figure 3e) and greatly exceed the 3-5 s averaging window used in gust time-scale calculations, and by the World Meteorological Organization and National Weather Service. Again consistent with previous research, $t_{rise}$ is best fit by a log-normal distribution (Table 2) (Hu et al., 2018). Short gust rise times are of particular interest in the wind energy industry, since gusts may ramp up faster than turbine pitch control systems, which may take several seconds to respond, can mitigate the induced loads (Burton et al., 2011;Kanev and van

Engelen, 2010). The 1st percentile rise times are uniformly near 4 s and are most accurately estimated using a lognormal distribution fit to the complete sample, although all distribution types produce conservative predictions (Figure 4c). Gust lapse times tend to be longer than rise times (by 30 to 40%; Figure 3f), and also conform to a log-normal distribution (consistent with (Hu et al., 2018); Table 2). The implied asymmetry in the temporal evolution of wind gusts indicates that the Mexican-hat form often assumed in the wind energy industry is not realized (Hu et al., 2018) which has relevance to power control from wind

turbines under high wind gust magnitudes (Gottschall and Peinke, 2007). Gust length scales tend to be higher for the ridge towers (with modal values of 200 m (which is similar to the height of the ridges above in the intervening valley), and values of up to 1200-1400 m; Figure 3g) than in data from the valley towers, and to conform to log-normal or gamma distributions with only a small difference in goodness-of-fit (NLL) between these two distribution types (Table 2).



The results from analyses of data collected in the complex terrain of Perdigão are thus internally consistent across towers in terms of the distributional form that best describes the gust samples and are also generally consistent with analyses of sonic anemometer data at 65 m a.g.l. collected in moderate complexity terrain (Hu et al., 2018). To the extent distribution types are uniform across the site, and consistent with previous work (as they are for $U_{gust}$, $a_{gust}$ $k_{peak}$ and all three time parameters), it is

reasonable to conclude that the best-fit distributions identified herein are effective for describing wind gusts in complex and moderately complex terrain, although, as noted above these best fit distributions can be non-conservative when estimating values in the distribution tail, as is the case with $U_{gust}$ and the Weibull distribution.

Joint distributions of $U_{gust}$ at different heights on the same meteorological mast (upper-left off-diagonal panels with green boxes in Figure 5, see other joint distributions of other parameters in Supplementary Information) indicate large ratios of major to

minor axes and thus a strong association of gust intensity across heights of 10-100 m a.g.l. The aspect ratio (ratio of major to minor axis) of the joint probabilities of gust intensity at different heights also exhibits a clear influence from vegetation at the lowest measurement height. For example, in joint distributions of $U_{gust}$ on Tower 29 the aspect ratio declines from 3.08 (between measurements at 20 m and 10 m a.g.l.) to 1.99 (between 100 m and 10 m a.g.l.), but above 10-m a.g.l. ranges from 9.66 (between 30 m and 20 m a.g.l.) to 3.79 (between 100 m and 20 m a.g.l.). Joint distributions of gust length scale at different heights ($L_{gust}$,

bottom-right off-diagonal panels in Figure 5) conversely indicate very low ratios of axes length and hence weaker coherence. Joint probabilities of wind gust magnitude ($U_{gust}$) and length scale ($L_{gust}$) at the same height (i.e. the diagonal in Figure 5) indicate moderate aspect ratios (1.6 to 1.8) and thus coherence (consistent with previous research (Doran and Powell, 1982)), but there is little systematic variation with height. Thus, while gust length scales do not exhibit similarity across heights, there is strong vertical coherence in the magnitude of wind gusts across the layer from 20-100 m a.g.l..

Joint distributions of gust magnitudes at 60 m a.g.l. at towers 20, 22, 34 and 37 (Supplementary Information) indicate very high aspect ratios for values from the three towers on the southwest ridge (7.8-8.4), but considerably lower values with data from tower 20 (within the valley, of approx. 3). This again reemphasizes that although the occurrence of individual wind gusts at towers along the ridge is not simultaneous (see conditional probabilities discussed below), the probability distributions of their magnitudes are similar. Conversely, consistent with results shown in Figure 5, joint distributions of gust length scales and time

scales across all towers indicate lower consistency (as manifest in smaller aspect ratios) (see Supplementary Information).



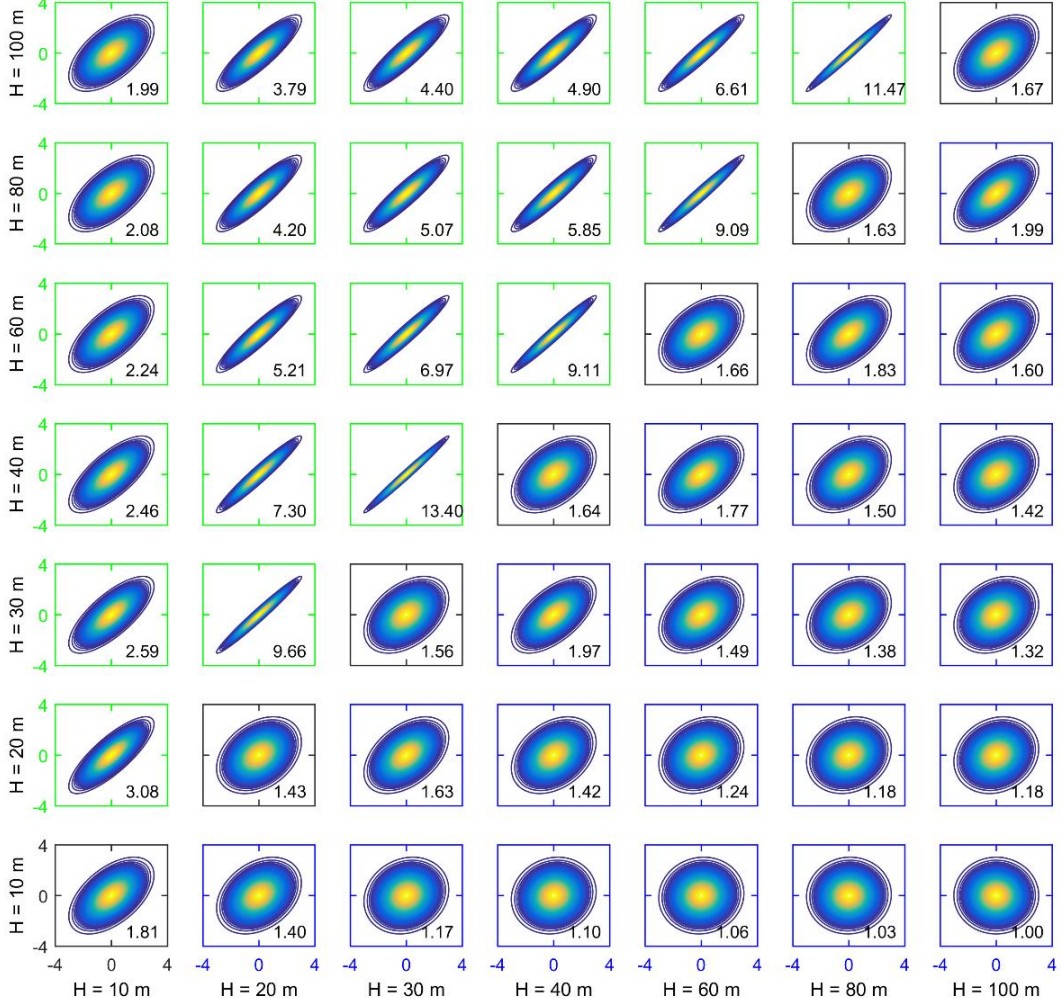

**Figure 5: Joint distributions of $U_{gust}$ and $L_{gust}$ at different heights (H) from Tower 29. Upper-left off-diagonal panels with green boxes show the joint distributions of $U_{gust}$ at each pair of heights. Diagonal panels with black boxes show the joint distributions of $U_{gust}$ and $L_{gust}$ at the same height. Bottom-right off-diagonal panels with blue boxes show the joint distributions of $L_{gust}$ at each pair of heights. The number at the bottom-right corner of each panel is the average ratio between the major axis and minor axis of ellipses. There are 99 ellipses corresponding to the confidence levels from 1% (the innermost ellipse with the warmest color) to 99% (the outermost ellipse with the coldest color).**

## 4.2   Conditional sampling of wind gust properties

Consistent with previous research, the probability of a wind gust varies systematically with dynamic stability and is higher under near-neutral and unstable conditions (Hart and Forbes, 1999), although gust probability at the valley towers is most similar to the ridge towers during times of very high (> 0.25) and very low (< 0.1) turbulence intensity (TI) (Figure 6, column 3). Large values of $U_{gust}$ tend to occur during stable and near-neutral conditions, especially at the ridge-top towers (Figure 6, column 1) (stable conditions are also found to be associated with higher $U_{mean}$). Consistent with previous research GF exhibit only a weak dependence on prevailing stability (Agustsson and Olafsson, 2004), but both GF and gust length scales scale with TI (Ashcroft, 1994;Greenway, 1979;Hu et al., 2018). Gust length scales are largest under the most unstable conditions particularly at the ridge towers (Figure 6), while gust amplitude shows large inter-tower variability particularly during very stable conditions potentially



reflecting the role of orographic wave breaking in inducing high magnitude gusts (Durran, 1990;Hertenstein and Kuettner, 2005). This is consistent with the finding that $U_{gust}$ at the ridge towers decrease with increasing TI (from 15 ms$^{-1}$ at low TI to 10 ms$^{-1}$ at TI > 0.3) since low TI is likely to occur under stable stratification when orographic forcing of standing waves is most likely to occur. Gust probability also shows a consistent dependence on wind direction and is highest for valley towers during

5      perpendicular flow (Figure 6, column 2), while under flow parallel to the ridges the empirical distributions of wind gust magnitudes, amplitudes and GF are rather similar for valley and ridge towers. This indicates that terrain-induced heterogeneity in wind gust characteristics is associated with perpendicular flow and that higher spatial resolution may be required to characterize wind gusts at sites with a high frequency of cross-ridge flow.

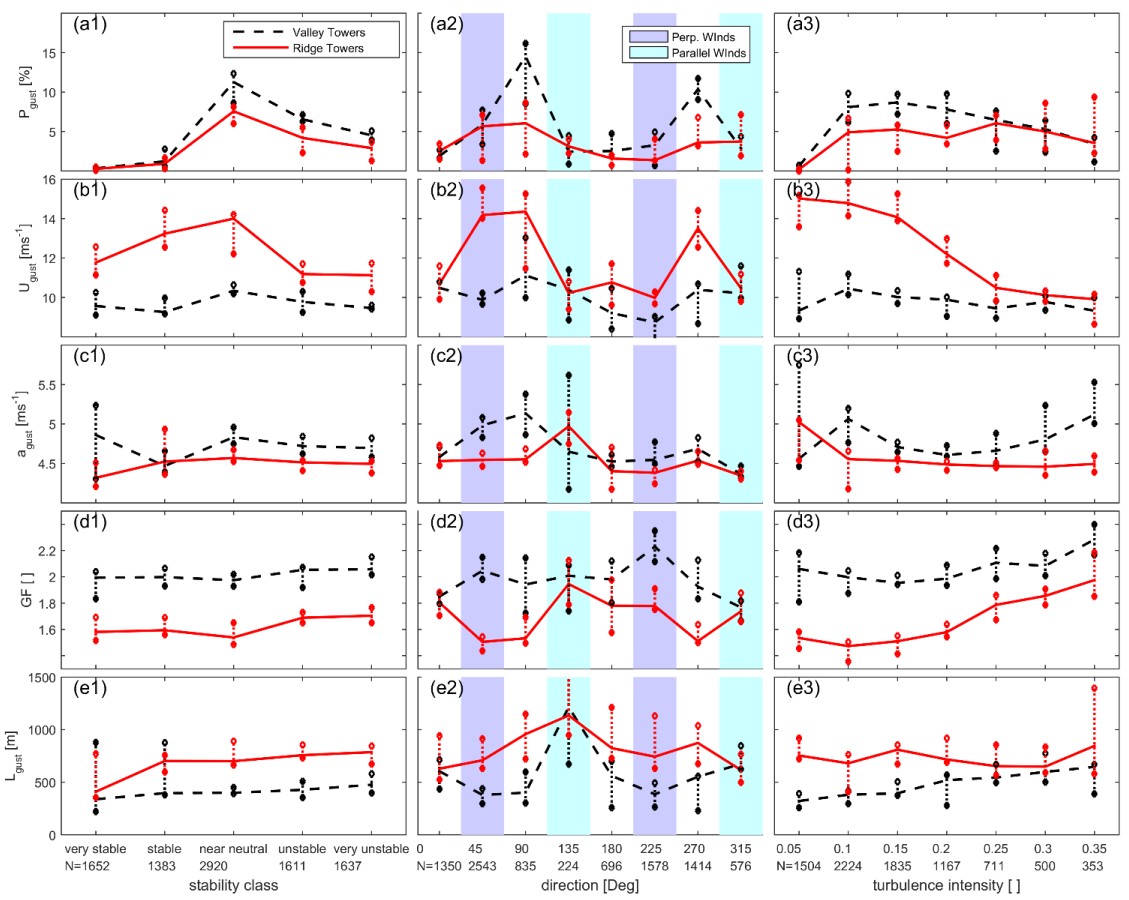

**Figure 6: Median gust parameters during gusty conditions (Umean > 3 ms$^{-1}$, a$_{gust}$ > 4 ms$^{-1}$) at 60 m a.g.l. from ridge (solid red) and valley (dashed black) towers. The data sampled are conditionally sampled by stability classes based on Monin-Obukhov length (column 1), wind direction (column 2) and turbulence intensity (column 3) as measured at Tower 29 (60 m a.g.l.). Vertical whiskers at each data point denote the range of median parameter values for all towers in each class. N values in the second row of x-axis labels are**

15   **the number of 10 minute periods in each sample. One gust parameter is shown per row: (a) Probability of a gust (i.e. percentage of 10 minute periods that meet the gust criteria), (b) Gust magnitude (U$_{gust}$), (c) gust amplitude (a$_{gust}$), (d) gust factor GF, (e) gust length scale (L$_{gust}$).**

Data from all 10 minute periods within the 64 days of data considered herein exhibit positive shear in median $U_{mean}$ values from heights > 20 m a.g.l.. When the data are conditionally sampled to select only periods when wind gusts occurred (i.e. $U_{mean}$ > 3

20   ms$^{-1}$ and gust amplitude > 4 ms$^{-1}$), median gust length scale, time scale and magnitude also all increase with height at both ridge



and valley towers (Figure 7). Median gust duration also increases modestly but in an approximately linear fashion with height from 61 s at 30 m to 69 s at 100 m a.g.l.. This equates to an increase of 8 s duration over a 70 m height interval (i.e. 0.11 sm$^{-1}$) which is approximately one-third of the value (0.35 sm$^{-1}$) found for flat terrain (Román, 2017). The increase of gust duration with height is larger (0.20 sm$^{-1}$) during periods with high intensity wind gusts (i.e. $a_{gust} > 4$ ms$^{-1}$) (Figure 7e). Median GF and gust

amplitudes decrease with height irrespective of whether the data meet the criteria $U_{mean} > 3$ ms$^{-1}$ and $a_{gust} > 4$ ms$^{-1}$ (i.e. a strong gust) or not. In the total data sample median GF decreases from 1.4 at 30 m to 1.3 at 100 m at Towers 20 and 29, while for samples restricted to gust events the median GF decreases from 1.55 to 1.5 over the same height range. The change of GF with height is thus smaller than over a flat, homogeneous grassland where GF decreased by 0.2 over a 90 m layer (Suomi et al., 2015;Shu et al., 2016). In contrast to the data from the other meteorological masts, gust amplitudes from Tower 25 tend to

increase with height consistent with its location in the lee of a ridge which causes a pronounced reduction in $U_{mean}$, but has a lesser impact on $U_{gust}$.

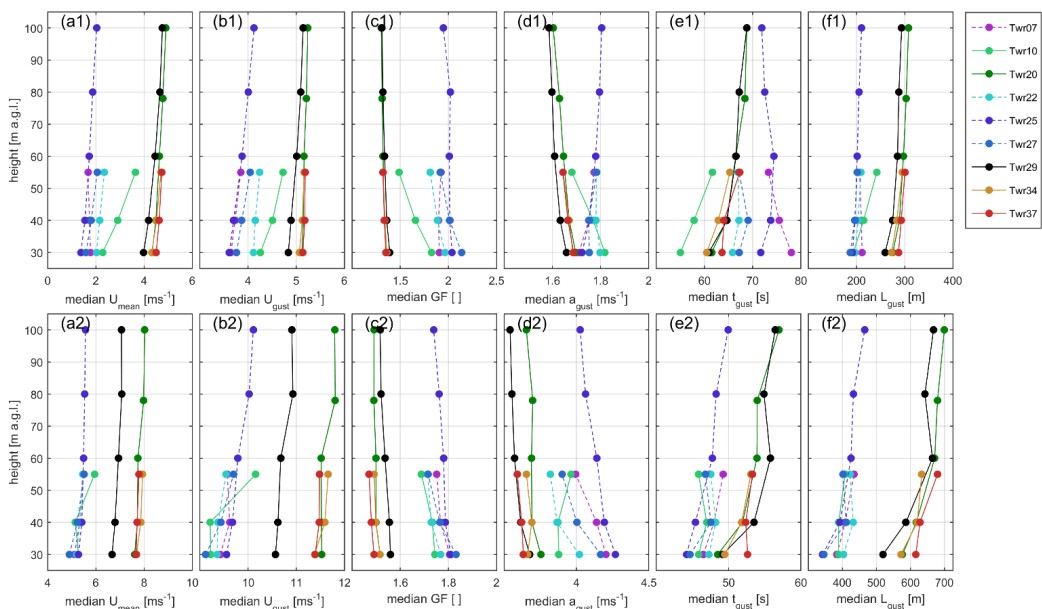

**Figure 7: Median values of the gust parameters, by height, at each of the nine towers. Row 1 shows median values computed for all 10 minute periods and row 2 shows median values for 10 minute periods which meet the gust criteria: $U_{mean} > 3$ ms$^{-1}$ and gust amplitude >**
**4 ms$^{-1}$. (a) 10 minute mean wind speed ($U_{mean}$), (b) 3 s maximum wind speed ($U_{gust}$), (c) gust amplitude ($a_{gust}$), (d) gust factor (GF), (e) gust duration ($t_{gust}$), (f) gust length scale ($L_{gust}$)**

Although the ZephIR lidar measurements are disjunct (at approx. 2 s) for each height and are subject to volumetric averaging, the probability distribution of $U_{gust}$ at 100 m a.g.l. derived from measurements with the ZephIR lidar (z423) located close to Tower 25 exhibits accord with that derived from the sonic anemometer deployed on this Tower (Figure 8a). The $U_{gust}$ distribution for

100 m a.g.l. from ZephIR z447 (which is at 74 m lower altitude and west of the SW ridge) indicates a much higher frequency of $U_{gust} < 3$ ms$^{-1}$ than at any of the 100 m towers or ZephIR z423 (Figure 8a) partly due to sheltering by the SW ridge under northeasterly flow. Vertical profiles of wind gust magnitudes and mean wind speeds follow a power law form under some circumstances (Brook and Spillane, 1970;Stull, 2012). Median gust factors (GF = $U_{gust}/U_{mean}$) from both the ZephIR lidars and sonic anemometers on the towers also conform to a power law with height (z), although the power law coefficients and quality of

the fit vary between sampling location and instrument. As shown in Figure 8b, the GF profiles are more linear below the





elevation of the ridge tops and data from the ZephIR lidars indicate a higher power law coefficient consistent with suppression of gust maxima from increased volumetric averaging with height (Suomi et al., 2017). The power law coefficients for the GF dependence on height are approx. -0.04 to -0.05 in data from the sonic anemometers deployed on the ridge-top towers, but is double that for the tower in the valley (25) and are -0.13 and -0.17 in data from the ZephIR lidars (Figure 8c). For comparison,

data from flat surfaces indicate lower GF values and smaller power law coefficients derived from GF profiles (-0.034 to -0.051, with larger magnitude values under more stable conditions) (Suomi et al., 2015) than those observed in the current study (Figure 8c).

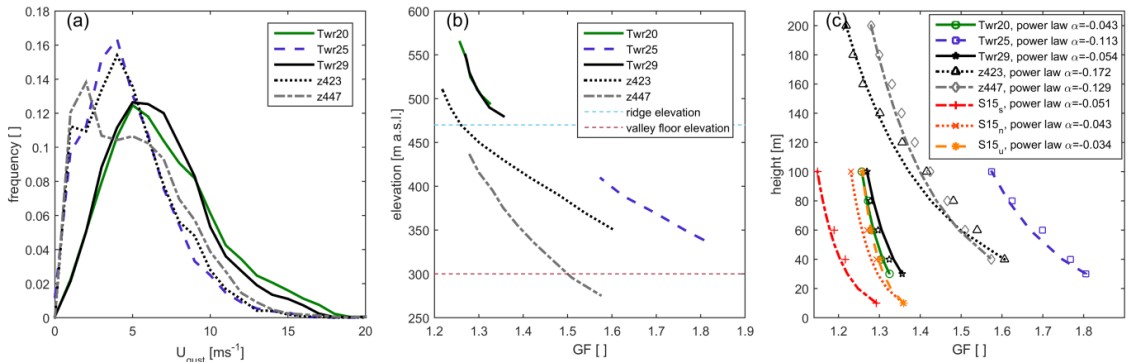

**Figure 8: (a) Probability distributions of gust magnitude ($U_{gust}$) at 100 m a.g.l. as derived from sonic anemometers deployed on the**
**three 100 m towers and from the ZephIR lidars. (b) Median GF versus height above sea level (a.s.l) for all 10 minute periods with $U_{mean}$ > 3 ms$^{-1}$ from sonic anemometers and ZephIR lidars. The ridge and valley floor elevations above sea level are shown by the horizontal dashed lines. (c) Power law fits to median GF as a functions of height a.g.l. for 10-minute periods with $U_{mean}$ > 3 ms$^{-1}$. Also shown are results from measurements over flat terrain (Suomi et al., 2015), under stable (S15$_s$) near-neutral (S15$_n$) and unstable (S15$_u$) conditions**

Gust peak factor ($k_{peak}$) derived from sonic anemometer measurements at 60 m a.g.l. greatly exceed 1.7 (Woetmann Nielsen and
Petersen, 2001) and results from the empirical expression in Eq (9) (Wieringa, 1973). Increasing the pre-factor in Eq. (9) from 1 to 2.4 (shown as Wieringa* in Figure 9) leads to a more conservative approximation, which exceeds 90% of observed $k_{peak}$ values for a given sustained wind speed, but the ratio of the 3 s gust amplitude to the standard deviation of the longitudinal wind speed is substantially enhanced in complex terrain and is not well described by either approximation.

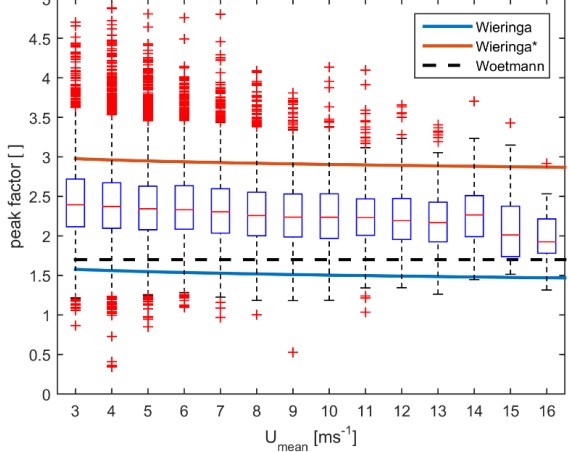

**Figure 9: Box plot of gust peak factor ($k_{peak}$) estimates at 60 m a.g.l. from all nine towers as a function of $U_{mean}$. Also shown are estimates from Eq. (9) (Wieringa, 1973) and a revised empirical function in which the leading term in Eq. (9) is increased from 1.0 to 2.4 (Wieringa*) and a constant value of 1.7 value (Woetmann Nielsen and Petersen, 2001).**



### 4.3 Spatial co-occurrence of wind gusts

The mean marginal gust probability at 60 m a.g.l. across all nine towers is 2.7% (range of 1.8% to 5.1%, Figure 10a). Thus, on average in any 10 minute period at any tower there is a 2.7% chance that $U_{mean} > 3$ ms$^{-1}$ and $a_{gust} > 4$ ms$^{-1}$. The mean wind gust co-occurrence probability at 60 m between towers has a mean value of 0.27 indicating that if a wind gust is detected at one given

tower in a 10 minute period there is a 27% chance that there will be a gust detected at another given tower in the same 10 minute period. When adjacent 10 minute periods are included the mean co-occurrence across towers rises to 0.42 (Figure 10a). The asymmetry in the conditional probabilities shown in Figure 10 reflects the fact that marginal probability of gusts varies across the study area. The towers are separated by distances of 221 to 1666 m (mean separation of 912 m), thus although the length scale analysis shown in Figure 3(g) indicates that individual wind gusts may not have a sufficient spatial scale to 'engulf' two towers,

the flow in which these features are embedded is likely to spawn multiple of these coherent transient features across the site. Towers 10 and 20 have largest overall conditional probability of gust co-occurrence with all other towers (Figure 10a) in part due to their high marginal wind gust probabilities (4.3 and 5.1%, respectively). Conversely, wind gusts at Towers 34 and 37 are not strongly associated with gusts at other towers especially those located within the valley (Figure 10a). Towers 20, 34 and 37 are all on the southwest ridge, and their heterogeneity in marginal (1.8 to 5.1%) and conditional probabilities (10 to 40%) of wind

gust occurrence indicate significant spatial variability in flow conditions and the presence of these intermittent coherent structures along the ridge. Indeed the mean gust co-occurrence among these three southwest ridge towers is lower than the site-wide mean. These wind gust co-occurrence probabilities are thus greatly distorted by terrain-flow interactions and are substantially smaller than those reported for flat terrain (of up to 90% for locations separated by 200 m (Branlard, 2009)).

At Towers 20, 25 and 29 (the three 100 m towers), the mean intra-tower gust co-occurrence probabilities (computed across

heights on the same tower) are 55, 60 and 55%, respectively. It is noteworthy that the two ridge-top towers have higher gust probabilities (both conditional and marginal) at their lowest measurement heights; 10 and 20 m, and these probabilities decrease strongly with height (Figure 10b,d). Conversely, marginal wind gust probabilities are highest at 80 and 100 m in data from Tower 25 (in the valley) and there is some evidence of a decoupling of data from this tower between heights above and below 80 m as manifest in high joint probabilities of gusts in data from 80 and 100 m, and between sonic anemometers at 30, 40 and 60 m

a.g.l., but low conditional probabilities between data collected at 80 m and, for example, 40 m.







**Figure 10: Co-occurrence of wind gusts in individual 10 minute periods between sonic anemometers deployed on different towers (frame a) or at different heights on the same meteorological mast (frame b-d). The color scale for each of the four panels indicates probability of a wind gust at anemometer 2 (on the vertical axis) during any 10 minute period that also has a gust at anemometer 1 (on the horizontal axis). The marginal probability of a wind gusts is also shown along the y axis. (a) Co-occurrence probabilities for gusts at 60 m a.g.l. across all 9 towers. Towers are sorted by their distance from Tower 29, colored dots in each square show the probability of co-occurrence when the time window is extended to 30 minutes. Also shown is the co-occurrence of wind gusts at different heights on (b) Tower 20, (c) Tower 25 and (d) Tower 29.**

## 4.4    Spectra and Coherences

Normalized power spectra of longitudinal wind speeds from the sonic anemometers during gust periods differ in three primary ways from the mean spectra derived as the composite of all two hour non-gust periods. Firstly, during gusty periods the spectral peak is shifted to the left (to lower normalized frequencies), second, the spectral peak is more distinct and lastly, the spectra exhibit lower variance at higher frequencies (normalized frequency > 1) (Figure 11). The relative magnitude of spectral peak at normalized frequencies of 0.07 to 0.17 is greater during gusty periods in data from the valley towers than the ridge towers (Figure 11a) and is in the same frequency range (i.e. period ≈ 120 s) as in less complex terrain (Hu et al., 2018). At the three tall

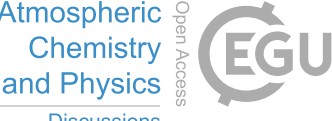



towers, the proportion of wind speed variance associated with normalized frequencies > 1 decreases monotonically with height, while the share of variance expressed at low frequency increases (Figure 11b-d). Composite spectra during stable and unstable conditions (Figure 11e) indicate that during stable conditions, the variance at both ridge and valley towers is shifted toward lower normalized frequencies in contrast to previous research in rolling terrain that indicated a shift of variance towards higher

normalized frequency during stable conditions (Panofsky et al., 1982). Composite spectra conditionally sampled by wind direction indicate that when the flow is parallel to the ridges (within ±15° of NW or SE), consistent with results shown in Figure 6, the power spectra of longitudinal winds from the ridge and valley towers are very similar (Figure 11f), while during periods of perpendicular flow the variance at the valley towers is shifted to higher normalized frequencies (consistent with the energy cascade induced by topographic forcing and previous work at lower measurements height (1.6 m) downstream of a smaller

obstruction (11.6 m) (Panofsky et al., 1982)).

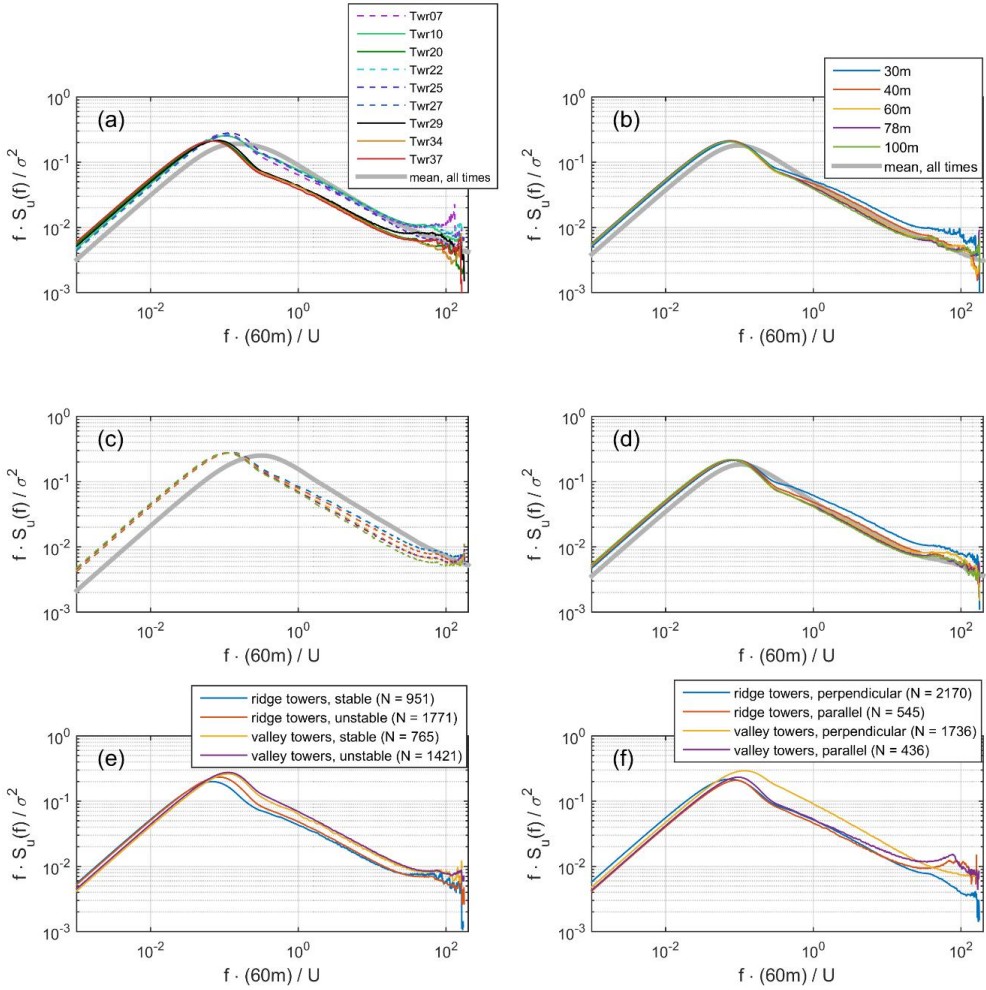

**Figure 11: Power spectra of wind speeds during gusty conditions ($U_{mean} > 3$ ms$^{-1}$ and $a_{gust} > 4$ ms$^{-1}$) (a) Spectra of wind speeds at 60 m a.g.l. on the 9 towers. Frames (b-d) show the normalized composite spectra from b) Tower 20, c) Tower 25 and d) Tower 29 from multiple heights. (a-d) also show the mean normalized spectra for all measurement periods (including non-gust periods). (e) Mean**
**normalized spectra at ridge and valley towers, conditionally sampled by stability class: stable (including very stable) and unstable (including very unstable) conditions and (f) mean normalized spectra at ridge and valley towers, conditionally sampled by wind direction: parallel (±15° of NW or SE) and perpendicular (±15° of NE or SW) to the ridges.**




Power spectra of horizontal wind speed computed for all ten ZephIR lidar measurement heights and from sonic anemometers deployed on the 100 m meteorological masts indicate relatively wide variability in the magnitude of the diurnal peak, $S_p$, from $0.6 \times 10^4$ to $2.6 \times 10^5$ m²s⁻¹ due to variations in surface forcing across the site and instrumentation differences. Preliminary estimates for the reverse height (i.e. the height of minimum variance at a frequency equal to 1 day⁻¹) derived from the ZephIR

lidars (z423 in the central valley and z447 that was deployed outside the ridge-valley system), are approximately 180 m a.g.l. (Figure 12) which is higher than reported for coastal sites (approx. 120 m a.g.l. (Larsén et al., 2018)). Reverse height estimates from sonic anemometers on the ridges (tower 20 and 29) and ZephIR lidar z423 are approximately 40-80 m above the ridge top (Figure 12).

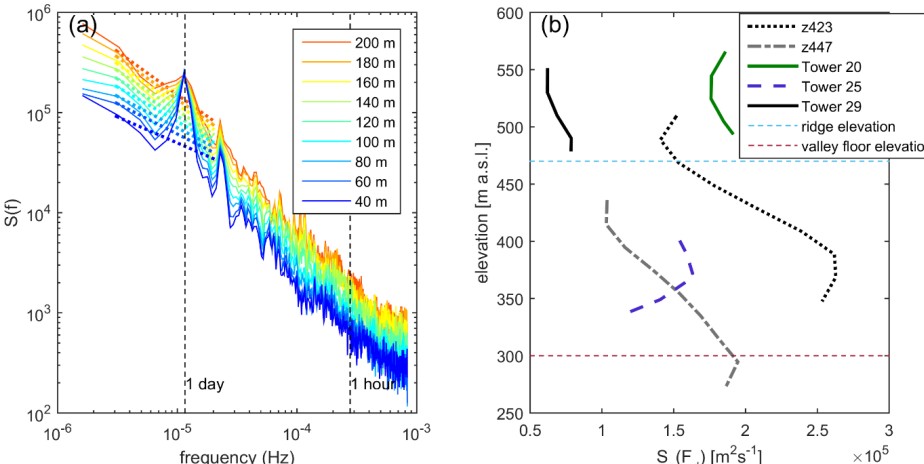

**Figure 12: (a)** Power spectra of horizontal wind speeds at ten heights from ZephIR lidar z447 and linear fits defining the trend in each near the frequency 1 day⁻¹. **(b)** Vertical profiles of the magnitude of the diurnal peak in wind speed variance $S_p$, from each ZephIR lidar and the sonic anemometers deployed on each 100 m tower. Measurement heights are shown by their elevation above sea level (a.s.l.). The towers and lidars are denoted by the color scheme introduced in Figure 1. Ridge and valley floor elevations are also shown.

Coherence functions of longitudinal wind speed from horizontally-separated sonic anemometer pairs do not conform to an

exponential form, and instead exhibit a marked concave-down section at reduced frequencies below 0.7 (Figure 13a). Nevertheless, functional values are substantially higher for ridge towers (excluding 10) than for valley towers, indicating greater coherence across the top of the valley than within the valley. Fitted C values (from Eq. 11) range from 1 at some ridge top towers up to 5 at tower 27 (one of the most sheltered towers). These C values, computed for horizontal separation, are toward the low end of C values reported in previous research that indicate C estimates between 5 and 15, and that for sensor separation distance

to height ratios > 2 (which is the case for inter-tower coherences in the current study), the mean decay rate was greater than 19 (Solari, 1987;Larsén et al., 2016). It is possible that the discrepancy in C values with previous research is due to poor fit to an exponential form in coherence functions (Figure 13a), although it may also reflect faster decoupling of flow regimes in complex terrain. Intra-tower coherences derived using data from different heights relative to data from 60 m a.g.l., are well described by Eq. (11) (when fitting the section above the coherence floor). Coherence functions for the 100 m ridge towers (represented by

Tower 29, Figure 13 b) have C values between 8.4 and 10. 6, which lie in the middle of the range for sensors with vertical separation of 6 to 16.9 (Solari, 1987). These values are higher overall than those measured for longitudinal separation, which is consistent with Taylor's Hypothesis: that turbulent structures evolve slowly, as they are advected by the mean flow (Larsén et al., 2016). Though Tower 25 had among the highest decay rate of coherence with the reference tower, the intra-tower coherences for Tower 25 exhibit C values of 4.6 to 6.7 (Figure 13c). Consistent with expectations, the sensors closest to 60 m a.g.l. at each



tower have the highest coherence with measurements at 60 m a.g.l., while the measurements at 30 m a.g.l. sensor (the lowest height included in this analysis) have the lowest coherence values (and highest decay rate). The vertical coherence functions from Perdigão towers decay to their minimum value at a reduced frequency of ~ 0.1, which is smaller than the value of 0.36 calculated for wind over water (Mehrens et al., 2016), and implies the terrain limits the size of coherent wind structures.

5   Coherence functions derived using wind speeds from ZephIR lidar z423 show slightly larger C values that are derived from sonic anemometer data from the nearby Tower 25, and exhibit high C values between measurements at 60 m a.g.l. and those above 100 m a.g.l. indicating a reduction in the degree to which flow at these heights is coupled (Figure 13d).

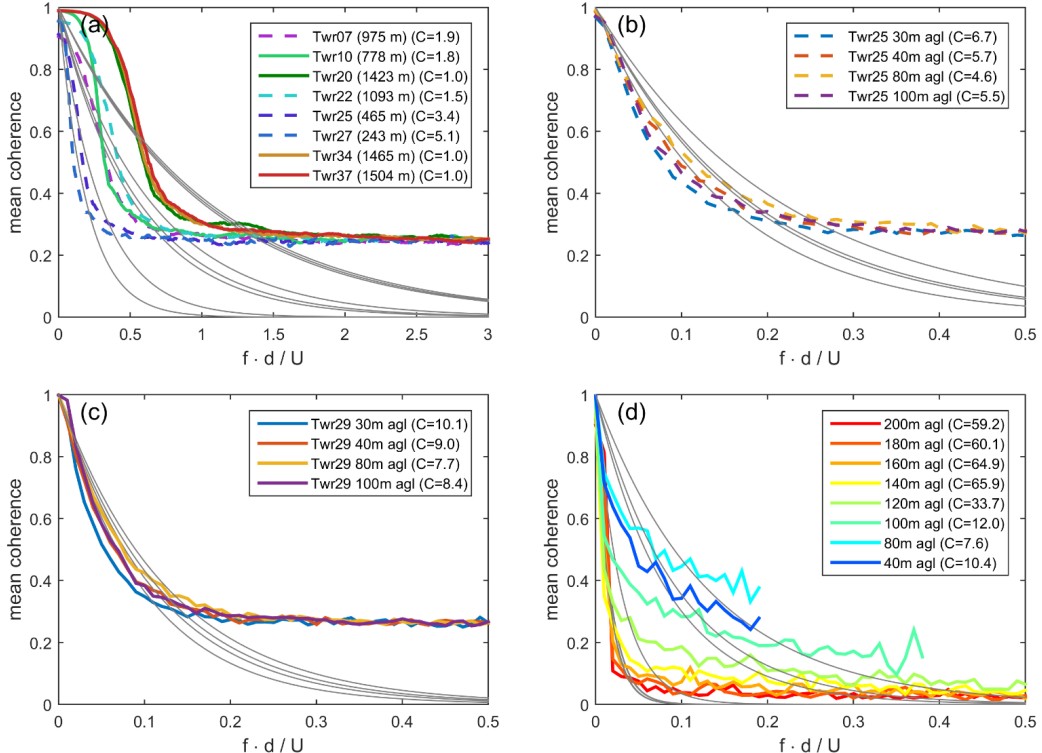

**Figure 13: Mean coherence in longitudinal wind speeds as measured by sonic anemometers based on 2 hour time series. (a) Mean coherences between Tower 29 and each other tower, for a 60 m measurement height, and exponential fits there to. (b) Coherences between each height and the 60 m reference height at Tower 20. (c) Coherences between each height and the 60 m reference height at Tower 25. (d) Coherence in horizontal wind speed at each height and the 60 m reference height measured by ZephIR lidar z423.**

## 5   Concluding Remarks

The experiment conducted at Perdigão provides an unprecedented data set for studying flow characteristics in complex terrain.

15   Herein we focus on wind gust characteristics as described using six months of data recorded at 18 Hz from 51 3-D sonic anemometers deployed on nine tall meteorological masts at heights of 10 to 100 m and two vertically pointing Doppler lidars. Consistent with previous research, analyses presented herein illustrate substantial spatial heterogeneity in the magnitude, scale and occurrence of wind gusts over an area of approximately 3 by 3 km (Figure 1) and reemphasize the complex effects of terrain forcing on near-surface flow.

20   Nine properties of wind gusts (intensity measures; magnitude, amplitude, peak factor and gust factor, and scale metrics; rise and lapse time, duration and length scales) exhibit similar parent probability distributions to those derived from measurements in





moderately complex terrain (Hu et al., 2018), indicating that these distributional forms may be generalizable. However, the best-fit distributional forms (selected using negative log-likelihood) underestimate the magnitude and amplitude of intense gusts (i.e. the 99[th] percentile values). Although the wind gust characteristics (including probability of gust occurrence) exhibit similar distributional forms across the site, they differ greatly in terms of the shape and scale parameters of the distributions as applied to

data from locations in the valley compared to the ridge tops. These differences are less pronounced when the flow is parallel to the ridge orientation. Gust length scales on the ridge tops are frequently similar to the height of the ridge above intervening valley (which is approx. 175 m deep), and the modal gust length scale measured in the valley is 200 m.

There are clear commonalities in gust properties across the site and between estimates derived using data from sonic anemometers and vertically-scanning Doppler lidar. Joint probability distributions of the gust properties indicate high aspect

ratios for gust intensity metrics across different measurement heights and locations. However, low aspect ratios are evident for wind gust length scales computed from sonic anemometers deployed at two heights on the same meteorological mast and/or from sonic anemometers deployed on meteorological masts separated by distances of 200 to 1600 m in the horizontal, and the co-occurrence probabilities of wind gusts across the site illustrate the very high complexity of flow over what is superficially a simple two-dimensional valley enclosed by two parallel ridges (Figure 1). These results further indicate that terrain features (and

the vegetation canopy) may have a more profound impact on the dimensions of wind gusts than their magnitude.

The amount of variance in wind speeds associated with the diurnal cycle varies depending on measurement system and location within the study site. Nevertheless, there is evidence that the reversal height (where the first order effects of heat exchange at the land surface is minimized) is ~ 60 m above the ridges that enclose the valley. This height is consistent with a decoupling of flow derived from the coherence functions estimated from the vertically scanning Doppler lidars (as indicated by the step change is C

values, Figure 13b), and thus it is postulated that this may also be representative of the height at which gust properties may also exhibit diminishing dependence on local surface characteristics.

Gust co-occurrences and coherence statistics indicate the presence large-scale gust phenomena that are simultaneously manifest at the ridge towers but not the valley towers. Gust occurrence across the Perdigão site is significantly influenced by the terrain resulting in much lower average gust co-occurrence probability (of 27%) across towers than those observed in flat terrain

(Branlard, 2009). The decay of coherence functions for vertical displacements is in the range found in flat terrain (Solari, 1987;Vigueras-Rodríguez et al., 2012). However, coherences for the large horizontal displacements (> 700m) between towers do not conform to an exponential fit (as with those presented in (Mehrens et al., 2016)) and are characterized by smaller decay coefficients than have been found in research conducted in less complex terrain.

Data collected during the Perdigão experiment and analyses presented herein provide a foundation for improved wind gust

characterization in complex terrain. These data also provide an unprecedented opportunity for detailed validation and verification of numerical wind flow models (Butler et al., 2015;Suomi and Vihma, 2018).

**Data availability**

All data analyzed herein are available for download from the New European Wind Atlas data portal (2018) hosted by the

University of Porto and accessible at; http://perdigao.fe.up.pt.

**Acknowledgments**

We thank the Perdigão research team, especially the scientists and technicians of the Technical University of Denmark (DTU), INEGI, University of Porto, and the National Center for Atmospheric Research (NCAR) for their excellent work and logistical



support during the Perdigão measurement campaign. We particularly acknowledge the leadership of Prof. J. Mann and Dr. E. Dellwik of DTU for provision of the tree height data. We gratefully acknowledge funding support from the U.S. National Science Foundation (1565505), and U.S. Department of Energy (DE-SC001643). We are grateful to the municipality of Vila Velha de Ródão, landowners who authorized installation of scientific equipment in their properties, the residents of Vale do Cobrão, Foz do Cobrão, Alvaiade, Chão das Servas and local businesses who kindly contributed to the success of the campaign. The space for the operational centre was generously provided by Centro Sócio-Cultural e Recreativo de Alvaiade in Vila Velha de Rodão.

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
