# Peer review of "Characterizing Wind Gusts in Complex Terrain"

_Atmospheric Chemistry and Physics, 2018_

## Referee Comment (RC1) · Anonymous Referee #1 · 24 Sep 2018

The manuscript deals with gusts in a very complex terrain – the Perdiago experiment. It is a very comprehensive data evaluation of high quality measurements mainly from 9 meteorological towers, but also includes measurements from two vertically pointing wind lidars. The analysis is very comprehensive and interesting. It covers the description of gusts broadly and not only limited to "3 sec meteorological gusts" or the "Mexican Hat" gusts for wind energy. In this respect it was a pleasure to read the manuscript. I have a few comments for the authors to consider: 1. Eq.(1). There seems to be a minus missing 2. Eqs. (6), (7) and 8(9. Is it possible to provide references for the transfer functions. 3. Although in the introduction the broad importance of gusts is described, on page 8 line 7 the thresholds that are applied in the data analysis here relates to wind turbines and wind energy. This limitation in the study should be mentioned more

specifically and put up-front. 4. Figure 5. It is difficult (impossible for me at least) to distinguish between the frame of the blue and black boxes. 5. Figure 10. The figure and its legend are very busy. I suggest having figure 10a as a separate figure. 6. Page 19, line 10 to page 21, line 14, I find the idea to devise specific spectra for gusts periods and non-gusts periods as rather artificial and it does not contribute much to the overall discussion on gusts. The same goes for the discussion on the reverse height, which also does not adhere to the findings of the gusts characteristics.

---

## Referee Comment (RC2) · Anonymous Referee #2 · 2 Nov 2018

General Comments

The manuscript entitled "Characterizing Wind Gusts in Complex Terrain" by Letson et al. addresses various aspects of wind gusts based on nine meteorological masts and two Doppler lidars within a region of about 3 km x 3 km in size, that is characterized by two topographic ridges separated by a valley, which are oriented in the northwest to southeast direction. Although it is not explicitly expressed in the manuscript, the study seems to be a follow-up of the work by the same authors (Hu et al., 2018), but here the attempt is to generalize the findings of Hu et al. (2018) by analyzing several mast measurements instead of one only. The present work focuses in understanding the statistical behavior of wind gusts as a function of height, in the horizontal plane, and as a function of atmospheric conditions (stability, wind direction, turbulence inten-

sity). The paper presents in-depth analysis of various gust parameters using different types of statistical methods. However, due to this, in many parts the paper is laborious to read and to understand. Therefore, it is strongly recommended to restructure the manuscript by focusing more on the presentation of the key objectives of the work and addressing the results and conclusions accordingly. In other words, substantial revision is recommended.

Specific Comments

Introduction (page 1, lines 24-37): It is not clear what you mean by a wind gust. You refer to gusts generated by mesoscale convective systems and downdrafts and by mountain waves, but is the topographic channeling also causing wind gusts? Recommendation: provide a clear definition for a wind gust already in the very beginning of the manuscript.

The use of terminology is confusing in some parts of the work:

- Gust definitions are provided only on page 6, and before that you use terminology "descriptors of wind gusts" (p. 3, line 1), "wind gust characteristics" (p. 3, line 33) and "gust parameters" (p. 4, lines 19-20), without any explanations. Recommendation: define the gust already in the beginning (with a reference to Section 3.1) and use the terminology consistent within the whole manuscript.

- Doppler lidar scanning technique: In the abstract and in different parts of the manuscript you write that the Doppler lidars are "vertically pointing", "vertically-pointing conically scanning" and "vertically-scanning", which is confusing. Moreover, in Section 2 (p. 5, lines 12-30) you indicate that the lidar is a scanning lidar with cone angle of +/-15°, but you don't provide any information on, e.g., how many lines-of-sight the Doppler lidar has, etc. Recommendation: provide detailed technical information of the Doppler lidars and the scanning techniques in Section 2. If the Doppler lidar was not applied in "stare mode" (i.e. measuring only in vertical direction) you can write "vertical profiles derived from Doppler lidar measurements" instead of "vertically pointing".

Objectives are expressed quite broadly in Section 1 (pages 3-5) and the conclusions in Section 5 do not include answers to all objectives. Recommendation: Provide a more concise list of objectives and check that you answer them in Section 5.

Section 2: Reorganize the Section: start with the most important data set (met masts), then Doppler lidars, and in the end other supporting data (terrain elevation + canopy height). Adding subtitles would make the Section more organized.

Figure 2c: percentages within each sector are difficult to read. Please, consider using bars instead of a line to indicate the histogram of wind directions. Section 3.1: Gust parameters were easier to understand after seeing the Figure 3a of Hu et al. (2018). Please, consider showing a similar picture also here. Furthermore, consider providing an equation for the gust length scale.

Section 3.1: Move the probability distributions (p. 6, line 34 – p. 8, line 5) into a separate subsection. Add the "gust period" (p. 8, lines 15-25) to the list of gust definitions on page 6. Consider also a new subsection for the gust parameterizations (p. 8, lines 6-14), or combine them to the list of gust parameters on page 6 (or consider leaving them out from the whole study).

Page 12, lines 10-11: comparison of 3-5 s averaging window and the gust rise time is unfair, as the gust rise time is defined based on 3 s moving averaged wind speed time series. Moreover, in Figure 4c, the 1% t_rise data are scattered around 4 s. Is this related to the presence of the lower limit for t_rise arising from its definition? Please, comment.

Page 12, line 18: At this point it was necessary to read the paper by Hu et al. (2018) in order to understand the work in this manuscript. Recommendation: mention already in the introduction, that this present paper is strongly based on the work by Hu et al. (2018). You may even list their main findings in the introduction of this paper and then build the objectives of the present work on top of these findings, by addressing what else will be done here. This way, it will be probably easier to improve the structure of

the present paper when everything is not presented as "new".

Figure 5 and Figures S1-S5: Could these results be shown as a Table? The images are not very informative. Without seeing aspect ratios, it is impossible to visually see the differences between the cases where the aspect ratio changes e.g. from 6.97 to 7.30 or from 1.18 to 1.32 or even to 1.60. Recommendation: show a couple of examples of these joint distributions in Section 3, where the methodology is described, to illustrate visually, how the aspect ratio is calculated. Then, in the results section, you could summarize all the results in a Table providing only the aspect ratios, maybe by grouping the results into high, medium and low values. This way, the main results from the Supplementary material could also be shown in the main manuscript, supporting the nice results on page 13 lines 20-25, which are relevant and a key part of this study.

Figure 7 on page 16 is very difficult to interpret, because all the lines are close together and panels are very small. What is actually the added value of the upper panels (a1-f1) compared to the lower ones (a2-f2)? Moreover, the main results of Figure 7c2 are shown also in Figures 8b-c. Recommendation: remove all other panels except Figures 7e2 and 7f2, and modify the text accordingly. Only these panels provide something new compared to the other Figures.

Page 16, lines 17-25, and p. 17, lines 1-4: you discuss the Doppler lidar volume averaging, but in Section 2 you don't describe in detail what is meant by the "volume" in case of the continuous wave ZephIR lidar (and how it differs from pulsed Doppler lidars). Please, provide explanations.

Figure 9: Already Hu et al. (2018) showed that the peak factor is not a function of mean wind speed. Why do you show this picture? Please, consider removing it. Comparison to other parameterizations is not very relevant for this work, maybe you could remove the parameterization aspect from the whole paper, and leave it open for future studies. There is now enough material already for one paper, even without the parameterization.

Section 4.3. and Figure 10:

- In the text, you introduce a new concept "mean marginal probability" without defining it. Please, explain.

- Figure 10 is extremely difficult to interpret: color scale is continuous, but the results are presented as boxes. It is very difficult to interpret the color of each box; please, use categories in the color scale, for example at 0.1 intervals of probability.

- Concerning panel (a), you conclude that the mean co-occurrence probability is 0.27 for 10 min data and 0.43 for 30 min data. This means, that in 73% and 57% cases the gusts do not occur simultaneously. Moreover, the Figure 10a illustrates the small differences in co-occurrence probabilities especially in the low range (blue colors). Is that really necessary? Recommendation: provide results only for probabilities > 0.5 (and with 0.1 intervals in color scale as suggested in the previous comment).

- The color scale in panel (a) for 10 min data is different from the scale of 30 min data, because it is possible to distinguish the circles in the diagonal. Please, comment this.

- What is causing the asymmetry across the diagonal in panels b-d? It is especially pronounced in panels (b) and (d).

- Overall, Figure 10 is extremely difficult to interpret. What are the parameters on each side of the Figure? Why do you give percentages only on the vertical axes? Why the results are not symmetric with respect to the diagonal? Have you calculated the average probabilities using data from the diagonals too? It is possible to find the answers to these questions based on the provided information but it takes a lot of time. Due to this complexity, it is foreseen that many readers will probably skip the Figure while reading the paper. Therefore, the value of the illustration is questionable. Consider removing the Figure, or simplify it substantially.

Figure 11: Why do you show separately gusty conditions and all cases? Why not to show separately gusty and non-gusty cases (i.e. no overlap of underlying data)? The contribution of these spectra to the understanding of the characteristics of wind gusts

is questionable, please, consider removing this Figure.

Figure 13 and text on p. 21, lines 17-29, and p. 22, lines 1-7: you have recognized a coherence floor, but still you perform exponential fits with respect to zero – why? What is the parameter "d" on the horizontal axes?

Section 5:

- Conclusions do not answer directly to the objectives of this study. Please, check and modify.

- The third paragraph (p.23, lines 8-15) is too complicated. It is impossible to identify to which parts of the results section these conclusions refer to.

Technical Comments

p. 3, lines 1-3: long and complicated sentence, please simplify.

p. 4, line 1: Repetition: "characteristics", "characterized"

p. 4, lines 3-5: Complicated sentence: "Horizontal coherence........across the study domain." Please, clarify.

p. 5, lines 1-2: information in parentheses not necessary, please remove.

p. 5 line 12: "Some analyses reported herein employ..." Quite arbitrary approach, please, be more specific.

p. 10, line 19: Please provide the ZephIR lidar sampling rate here.

p. 12, line 17: Add: "Figure 3f-g".

p. 12, line 18: Remove parentheses: "with (Hu et al., 2018); Table 2", i.e., change to "with Hu et al. (2018); their Table 2".

p. 12, lines 18-20: Does this sentence summarize the results of the present study or those from the literature? Please, clarify.

p. 13, lines 3-7: Too complicated and long sentence. Please, simplify. Explain also what is meant by "all three time parameters".

p. 13, line 11: Please, define "gust intensity".

p. 13, line 14: Consider splitting the paragraph before "Joint distributions of gust..."

p. 15, line 18: Consider starting the sentence with "In Figure 7, data from all 10 minute periods within 64 days..."

p. 16, line 25: Add to the end of the sentence: "...the fit vary between sampling location and instrument (Figure 8c)."

p. 18, line 9: Is the referred Figure 3g correct here? Figure 3g shows the distribution of lapse time, not gust length scale.

p. 21, line 6: Should it read "Reversal height estimates"?

p. 21, line 25: "Tower 29, Figure 13b) have C values between 8.4 and 10.6" - is this wrong? According to caption, Figure 13b shows the results for Tower 20, and in panel (b) the C values are smaller.

p. 21, lines 25-26: "sensors with vertical separation of 6 to 16.9 (Solari, 1987)" - this is not understood: what is the "vertical separation" here? Does it refer to the instruments here or in the study by Solari (1987)?

p. 21, line 29: "Tower 25 exhibit C values of 4.6 to 6.7 (Figure 13c)" - these are probably also wrong. Please, check to be consistent with Figure 13.

p.23, lines 9-10: "Joint probability distributions of the gust properties indicate high aspect ratios for gust intensity metrics..." - What is the difference between the "gust properties" and "gust intensity metrics"? Please, explain.

p. 23, lines 10-14: the sentence starting from "However, low aspect ratios..." and ending to "...by two parallel ridges (Figure 1)." is very difficult to understand. Please,

simplify.

p. 23, line 19: Should it read "change in C"?

p. 23, line 20: Should it be "Figure 13d"?

———————————————————

---

## Author Comment (AC1) · 7 Dec 2018

Please find the authors' response to Anonymous Referees 1 and 2, and the revised manuscript with tracked changes, in the PDF associated with this comment. Thank you.
* * *

---

## Author Comment (AC2) · 7 Dec 2018

The manuscript deals with gusts in a very complex terrain – the Perdiago experiment. It is a very comprehensive data evaluation of high quality measurements mainly from

9 meteorological towers, but also includes measurements from two vertically pointing wind lidars. The analysis is very comprehensive and interesting. It covers the description of gusts broadly and not only limited to "3 sec meteorological gusts" or the "Mexican Hat" gusts for wind energy. In this respect it was a pleasure to read the manuscript. I have a few comments for the authors to consider:

**Thank you for your thorough and thoughtful response to this paper. Before addressing your specific suggestions, it is important to note that, at the recommendation of a researcher who contacted us after reading our submitted paper online, we are now using an improved method of estimating wind speed coherences. Since the imaginary part of the complex-valued cross spectral density function will tend toward zero as the data length used in the estimation increases, and its presence tends to bias the coherence floor upward. We have elected to use co-coherence as unbiased estimate of coherence as in Eliassen and Obhrai (2016). This has lowered the coherence floor in our estimates to near zero, and improved the exponential fits to coherence decay. This change is noted and explained in the text (Page 11, lines 13 – 19).**

1. Eq.(1). There seems to be a minus missing

**Thank you for noticing this error. It has been corrected**

2. Eqs. (6), (7) and (8) (9). Is it possible to provide references for the transfer functions?

**A reference has been provided for the four probability distributions (page 7 line 36)**

3. Although in the introduction the broad importance of gusts is described, on page 8 line 7 the thresholds that are applied in the data analysis here relates to wind turbines and wind energy. This limitation in the study should be mentioned more specifically and put up-front.

**While the threshold magnitudes chosen in this study are related to wind energy, the application of gust thresholds is necessary to the study of gust behavior, since mean-wind-speed-normalized parameters (such as gust factor, peak factor, and turbulence intensity) have extremely high variance at the lowest wind speeds. While the authors do not consider these thresholds to be a limitation, we now specifically describe the**

**threshold in the context of links to NWS (meteorological) protocols and the wind energy industry (Page 7, Lines 20 – 23)**

4. Figure 5. It is difficult (impossible for me at least) to distinguish between the frame of the blue and black boxes.

**The black boxes in Figure 5 (and the similar figures in supplementary materials) have been replaced with red, which should be easier for readers to distinguish from black.**

5. Figure 10. The figure and its legend are very busy. I suggest having figure 10a as a separate figure.

**Figure 10a and the color legend have been significantly increased in size (and panel b excluded). The color scheme has been changed to 10 categories rather than a continuous scale for increased clarity.**

6. Page 19, line 10 to page 21, line 14, I find the idea to devise specific spectra for gusts periods and non-gusts periods as rather artificial and it does not contribute much to the overall discussion on gusts. The same goes for the discussion on the reverse height, which also does not adhere to the findings of the gusts characteristics.

**Turbulence intensity (as a function of height) has been added to Figure 7, and a more explicit connection has been made between gust parameters (including TI) and reverse height (Page 17 lines 7-9) and (Page 26, lines 28-32)**

**Explanatory text has been added to elucidate the physical meaning of the gust and non-gust spectra and their relationship to gust phenomena (Page 25, lines 35 - 39)**

**Anonymous Referee #2**

General Comments

The manuscript entitled "Characterizing Wind Gusts in Complex Terrain" by Letson et al. addresses various aspects of wind gusts based on nine meteorological masts and two Doppler lidars within a region of about 3 km x 3 km in size, that is characterized by two topographic ridges separated by a valley, which are oriented in the northwest to southeast direction. Although it is not explicitly expressed in the manuscript, the study seems to be a follow-up of the work by the same authors (Hu et al., 2018), but here the attempt is to generalize the findings of Hu et al. (2018) by analyzing several mast measurements instead of one only. The present work focuses in understanding the statistical behavior of wind gusts as a function of height, in the horizontal plane, and as a function of atmospheric conditions (stability, wind direction, turbulence intensity). The paper presents in-depth analysis of various gust parameters using different types of statistical methods. However, due to this, in many parts the paper is laborious to read and to understand. Therefore, it is strongly recommended to restructure the manuscript by focusing more on the presentation of the key objectives of the work and addressing the results and conclusions accordingly. In other words, substantial revision is recommended.

**Thank you for your thorough and thoughtful response to this paper. Before addressing your specific suggestions, it is important to note that, at the recommendation of a researcher who contacted us after reading our submitted paper online, we are now using an improved method of estimating wind speed coherences. Since the imaginary part of the complex-valued cross spectral density function will tend toward zero as the data length used in the estimation increases, and its presence tends to bias the coherence floor upward. We have elected to use co-coherence as unbiased estimate of coherence as in Eliassen and Obhrai (2016). This has lowered the coherence floor in our estimates to near zero, and improved the exponential fits to coherence decay. This change is noted in the text (Page 11, lines 13 – 19).**

**Specific Comments**

**Introduction** (page 1, lines 24-37): It is not clear what you mean by a wind gust. You refer to gusts generated by mesoscale convective systems and downdrafts and by mountain waves, but is the topographic channeling also

causing wind gusts? Recommendation: provide a clear definition for a wind gust already in the very beginning of the manuscript.

**Wind gusts are now more clearly defined in the introduction as "coherent short-term wind speed maxima" (Page 1 line 32).**

The use of terminology is confusing in some parts of the work:

- Gust definitions are provided only on page 6, and before that you use terminology "descriptors of wind gusts" (p. 3, line 1), "wind gust characteristics" (p. 3, line 33) and "gust parameters" (p. 4, lines 19-20), without any explanations. Recommendation: define the gust already in the beginning (with a reference to Section 3.1) and use the terminology consistent within the whole manuscript.

**The wording of references to wind gust parameters has been changed to be more consistent. The above phrases have all been changed to 'wind gust parameters' since they refer to section 3.1**

- Doppler lidar scanning technique: In the abstract and in different parts of the manuscript you write that the Doppler lidars are "vertically pointing", "vertically-pointing conically scanning" and "vertically-scanning", which is confusing. Moreover, in Section 2 (p. 5, lines 12-30) you indicate that the lidar is a scanning lidar with cone angle of +/-15_, but you don't provide any information on, e.g., how many lines-of-sight the Doppler lidar has, etc. Recommendation: provide detailed technical information of the Doppler lidars and the scanning techniques in Section 2. If the Doppler lidar was not applied in "stare mode" (i.e. measuring only in vertical direction) you can write "vertical profiles derived from Doppler lidar measurements" instead of "vertically pointing".

**A detailed description of ZephIR lidar operation has been added do address these questions (Page 5, line 10 – Page 6, line 3)**

- Objectives are expressed quite broadly in Section 1 (pages 3-5) and the conclusions in Section 5 do not include answers to all objectives. Recommendation: Provide a more concise list of objectives and check that you answer them in Section 5.

**The conclusions section has been reorganized to include 5 bulleted paragraphs corresponding to the 5 bulleted paragraphs outlining the goals in the introduction (Page 2, line 6 to Page 4, line 5 and Page 24, line 13 to Page 25, line 30)**

**Section 2:** Reorganize the Section: start with the most important data set (met masts), then Doppler lidars, and in the end other supporting data (terrain elevation + canopy height). Adding subtitles would make the Section more organized. Figure 2c: percentages within each sector are difficult to read. Please, consider using bars instead of a line to indicate the histogram of wind directions.

**Thank you for this recommendation. Sub-sections have been added to section 2 in the suggested fashion. Figure 2 has been amended to include bars for the wind direction histogram in the wind rose panel.**

**Section 3.1:** Gust parameters were easier to understand after seeing the Figure 3a of Hu et al. (2018). Please, consider showing a similar picture also here. Furthermore, consider providing an equation for the gust length scale.

**Rather than adding an additional figure, explanatory text has been added to more clearly describe the definitions of rise time, lapse time and length scale (Page 6, line 28 to Page 7, line 3) and an equation has been added for length scale (Eq. 1)**

Section 3.1**:** Move the probability distributions (p. 6, line 34 – p. 8, line 5) into a separate subsection. Add the "gust period" (p. 8, lines 15-25) to the list of gust definitions on page 6. Consider also a new subsection for the gust parameterizations (p. 8, lines 6-14), or combine them to the list of gust parameters on page 6 (or consider leaving them out from the whole study).

**A new sub-section has been added for probability distributions and joint distributions.**

**The definition of gust periods has been moved to the end of this section (page 7, lines 13-22).**

Page 12, lines 10-11: comparison of 3-5 s averaging window and the gust rise time is unfair, as the gust rise time is defined based on 3 s moving averaged wind speed time series. Moreover, in Figure 4c, the 1% t_rise data are scattered around 4 s. Is this related to the presence of the lower limit for t_rise arising from its definition? Please, comment.

**This comparison is now explained more clearly and the presence of a lower limit on rise and lapse times resulting from the 3 s averaging window is made explicit (Page 13, lines 19-23)**

Page 12, line 18: At this point it was necessary to read the paper by Hu et al. (2018) in order to understand the work in this manuscript. Recommendation: mention already in the introduction, that this present paper is strongly

based on the work by Hu et al. (2018). You may even list their main findings in the introduction of this paper and then build the objectives of the present work on top of these findings, by addressing what else will be done here. This way, it will be probably easier to improve the structure of the present paper when everything is not presented as "new".

**We have added a more explicit statement that the current study is designed to mirror Hu et al., (2018) for ease of comparison to results from less complex terrain (Page 2, lines 12-14)**

Figure 5 and Figures S1-S5: Could these results be shown as a Table? The images are not very informative. Without seeing aspect ratios, it is impossible to visually see the differences between the cases where the aspect ratio changes e.g. from 6.97 to 7.30 or from 1.18 to 1.32 or even to 1.60. Recommendation: show a couple of examples of these joint distributions in Section 3, where the methodology is described, to illustrate visually, how the aspect ratio is calculated. Then, in the results section, you could summarize all the results in a Table providing only the aspect ratios, maybe by grouping the results into high, medium and low values. This way, the main results from the Supplementary material could also be shown in the main manuscript, supporting

the nice results on page 13 lines 20-25, which are relevant and a key part of this study.

**Figure 5 and the similar supplementary figures communicate more information than could be included in a table in a similar amount of space, and may be quicker to interpret. We now explicitly refer to each figure in SI and present in more detail the results shown there in. (Page 14 lines 15-37)**

**The description of the method used to compute the joint distributions has also been made more explicit. (Page 8, line 34 – Page 9 line 13)**

Figure 7 on page 16 is very difficult to interpret, because all the lines are close together and panels are very small. What is actually the added value of the upper panels (a1-f1) compared to the lower ones (a2-f2)? Moreover, the main results of Figure 7c2 are shown also in Figures 8b-c. Recommendation: remove all other panels except Figures 7e2 and 7f2, and modify the text accordingly. Only these panels provide something new compared to the other Figures.

**Half of the panels (those representing all 10 minute periods, rather than gust periods alone) have been removed from Figure 7. This makes the remaining panels much larger and easier to read.**

Page 16, lines 17-25, and p. 17, lines 1-4: you discuss the Doppler lidar volume averaging, but in Section 2 you don't describe in detail what is meant by the "volume" in case of the continuous wave ZephIR lidar (and how it differs from pulsed Doppler lidars). Please, provide explanations.

**A parenthetical explanation of the volume has been added "(the volume of the annulus swept out by the lidar beam, in a cone 30° from vertical)" (Page 17, line 15). This volume averaging in now also described on Page 5 (lines 26 – 31)**

Figure 9: Already Hu et al. (2018) showed that the peak factor is not a function of mean wind speed. Why do you show this picture? Please, consider removing it. Comparison to other parameterizations is not very relevant for this work, maybe you could remove the parameterization aspect from the whole paper, and leave it open for future studies. There is now enough material already for one paper, even without the parameterization.

**The authors believe that Figure 9 represents an important link to the meteorological literature and have decided to continue to include it in the paper. Hu et al. (2018) does show a weak relationship between peak factor and mean wind speed. Which has been found in the current study to be stronger in complex terrain.**

**Section 4.3. and Figure 10:**

- In the text, you introduce a new concept "mean marginal probability" without defining it. Please, explain.

**Some explanatory text for mean marginal gust probability has been added (Page 19, line 6)**

- Figure 10 is extremely difficult to interpret: color scale is continuous, but the results are presented as boxes. It is very difficult to interpret the color of each box; please, use categories in the color scale, for example at 0.1 intervals of probability.

**Figure 10a and the color legend have been significantly increased in size (and panel b excluded). The color scheme has been changed to 10 categories rather than a continuous scale for increased clarity.**

- Concerning panel (a), you conclude that the mean co-occurrence probability is 0.27 for 10 min data and 0.43 for 30 min data. This means, that in 73% and 57% cases the gusts do not occur simultaneously. Moreover, the Figure 10a illustrates the small differences in co-occurrence probabilities especially in the low range (blue colors). Is that really necessary? Recommendation: provide results only for probabilities > 0.5 (and with 0.1 intervals in color scale as suggested in the previous comment).

**Since, for a significant fraction of tower pairs, the co-occurrence is below 50% (as you note), we have elected to include the values for co-occurrences below this level. We believe that the binning of co-occurrence data into 10% categories (as you suggested above) has made these values easier to for the eye to interpret.**

- The color scale in panel (a) for 10 min data is different from the scale of 30 min data, because it is possible to distinguish the circles in the diagonal. Please, comment this.

**With an adjustment to make the color scale more precise, this artifact disappeared. Thank you for noting it.**

- What is causing the asymmetry across the diagonal in panels b-d? It is especially pronounced in panels (b) and (d). Overall, Figure 10 is extremely difficult to interpret. What are the parameters on each side of the Figure? Why do you give percentages only on the vertical axes?

**The marginal probabilities of gust occurrence listed along the vertical axis would be identical to those along the horizontal axis, since they only reflect the probability of gusts at each individual sensor. The asymmetry is largely due to the differing marginal probability of gusts at different sensors. A specific example of this asymmetry is called out and interpreted on Page 19 (lines 12 – 16).**

Why the results are not symmetric with respect to the diagonal? Have you calculated the average probabilities using data from the diagonals too? It is possible to find the answers to these questions based on the provided information but it takes a lot of time. Due to this complexity, it is foreseen that many readers will probably skip the Figure while reading the paper. Therefore, the value of the illustration is questionable. Consider removing the Figure, or simplify it substantially.

**See previous response on asymmetry. The figure has been simplified in the specific ways that the reviewer has suggested (enlarging of panel a and the use of probability classes rather than a continuous scale)**

Figure 11: Why do you show separately gusty conditions and all cases? Why not to show separately gusty and non-gusty cases (i.e. no overlap of underlying data)? The contribution of these spectra to the understanding of the characteristics of wind gusts is questionable, please, consider removing this Figure.

**The spectra for all periods are included to allow for direct comparisons with spectra from terrain of varying complexity in the literature (e.g. Panofsky et al. 1982 QJRMS)**

Figure 13 and text on p. 21, lines 17-29, and p. 22, lines 1-7: you have recognized a coherence floor, but still you perform exponential fits with respect to zero – why? What is the parameter "d" on the horizontal axes?

**At the suggestion of helpful commenter, we are now using the co-coherence as a non-biased estimator of the wind speed coherences. The coherence floor is no longer a major factor (see text Page 11, line 13 – 19)**

**Section 5:**

- Conclusions do not answer directly to the objectives of this study. Please, check and

modify.

**The conclusions section has been modified to have a structure parallel to the goals laid out in the introduction.**

The third paragraph (p.23, lines 8-15) is too complicated. It is impossible to identify to which parts of the results

section these conclusions refer to.

**The concluding remarks have been substantially reorganized. There are now five bulleted paragraphs (Page 24, line 13 to Page 25, line 39) corresponding to the bulleted goals listed in the introduction. Thoughts synthesizing information from more than one of the goals have been moved to the end of the section**

**Technical Comments**

p. 3, lines 1-3: long and complicated sentence, please simplify.

**This sentence has been simplified by removing the references to specific gust parameters (Page 3, Line 6-7)**

p. 4, line 1: Repetition: "characteristics", "characterized"

**Wording changed to avoid repetition (Page 4, Lines 1-2)**

p. 4, lines 3-5: Complicated sentence: "Horizontal coherence. . . . . .across the study domain." Please, clarify.

**This sentence was quite unclear. It has been re-written (Page 4, Lines 3 -5)**

p. 5, lines 1-2: information in parentheses not necessary, please remove.

**This information has been removed (Page 4, Lines 28-29)**

p. 5 line 12: "Some analyses reported herein employ. . ." Quite arbitrary approach, please, be more specific.

**This has been re-written to be more specific (Page 5, Line 11-15)**

p. 10, line 19: Please provide the ZephIR lidar sampling rate here.

**This has been added (Page 11, line 23)**

p. 12, line 17: Add: "Figure 3f-g".

**Added. (Page 13, Line 22)**

p. 12, line 18: Remove parentheses: "with (Hu et al., 2018); Table 2", i.e., change to "with Hu et al. (2018); their Table 2".

**Done. (Page 14, Line 1)**

p. 12, lines 18-20: Does this sentence summarize the results of the present study or those from the literature? Please, clarify.

**This has been reworded to indicate that it is a statement about the current study, which has been noted before in literature (Page 14, Lines 1-4)**

p. 13, lines 3-7: Too complicated and long sentence. Please, simplify. Explain also what is meant by "all three time parameters".

**This sentences has been divided into 2 sentences to make it easier to parse. The 'time parameters' have been listed explicitly (Page 14, Lines 10-14)**

p. 13, line 11: Please, define "gust intensity".

**Changed wording to "gust magnitude" which has been explicitly defined (Page 14, Line 15)**

p. 13, line 14: Consider splitting the paragraph before "Joint distributions of gust..."

**Done (Page 14 Line 23)**

p. 15, line 18: Consider starting the sentence with "In Figure 7, data from all 10 minute periods within 64 days. . ."

**This change has been made (Page 16 Line 18)**

p. 16, line 25: Add to the end of the sentence: ". . .the fit vary between sampling location and instrument (Figure 8c)."

**Reference added (Page 18, Line 6)**

p. 18, line 9: Is the referred Figure 3g correct here? Figure 3g shows the distribution of lapse time, not gust length scale.

**Thank you. This has been corrected to Figure 3i. (Page 19 Line 17)**

p. 21, line 6: Should it read "Reversal height estimates"

**Our preferred term in 'reverse height' the phrase 'reversal height' has been replaced throughout the paper for consistency.**

p. 21, line 25: "Tower 29, Figure 13b) have C values between 8.4 and 10.6" - is this wrong? According to caption, Figure 13b shows the results for Tower 20, and in panel (b) the C values are smaller.

**Thank you. The C values now changed due to the use of Co-coherence as a coherence estimator, mentioned above.**

p. 21, lines 25-26: "sensors with vertical separation of 6 to 16.9 (Solari, 1987)" - this is not understood: what is the "vertical separation" here? Does it refer to the instruments here or in the study by Solari (1987)?

**This has been re-worded for clarity (Page 23, lines 16-18)**

p. 21, line 29: "Tower 25 exhibit C values of 4.6 to 6.7 (Figure 13c)" - these are probably also wrong. Please, check to be consistent with Figure 13.

**Thank you. This has been corrected, and the values have changed somewhat due to the lower coherence floor associated with using co-coherence as an estimator.**

p.23, lines 9-10: "Joint probability distributions of the gust properties indicate high aspect ratios for gust intensity metrics. . ." - What is the difference between the "gust properties" and "gust intensity metrics"? Please, explain.

**These term have been replaced by 'gust parameters' and a specific example ($U_{gust}$), for consistency and clarity (Page 25, lines 1-2)**

p. 23, lines 10-14: the sentence starting from "However, low aspect ratios. . ." and ending to "...by two parallel ridges (Figure 1)." is very difficult to understand. Please, simplify.

**This has been separated into 2 sentences and simplified (Page 25, lines 2-4)**

p. 23, line 19: Should it read "change in C"?

**Yes, thank you. This has been corrected (Page 25 line 37)**

p. 23, line 20: Should it be "Figure 13d"?

**It absolutely should. This has been corrected (Page 25, line 37)**

[revised manuscript text omitted]